# Stop When Further Reasoning Won't Help:
# Attention-State Adaptive Generation in Reasoning Models

**Jiakai Li** [1]  **Ke Qin** [1 2]  **Rongzheng Wang** [1]  **Yizhuo Ma** [1]  **Qizhi Chen** [1]  **Muquan Li** [1]  **Shuang Liang** [1 2]

## Abstract

By incorporating test-time compute scaling, large reasoning models (LRMs) can solve complex problems through explicit chain-of-thought (CoT) reasoning processes. However, they often suffer from overthinking, resulting in redundant token outputs and degraded accuracy. Current methods to mitigate this issue remain limited: training-based approaches require substantial computational resources, while training-free methods rely on well-crafted prompts or unreliable confidence signals. In this work, we investigate early stopping from the perspective of attention distributions and propose a simple method, ASAG, which infers the model's reasoning state and adaptively adjusts the generation strategy. The proposed framework is training-free and plug-and-play, enabling seamless integration into existing LRMs. Extensive experiments on nine benchmarks demonstrate consistent improvements across mainstream LRMs with varying parameter scales, including the DeepSeek-R1-Distill and Qwen3 series. Specifically, ASAG improves average accuracy by 3.2% while reducing the number of generated tokens by nearly 40% across all reasoning tasks on Qwen3-8B.

## 1. Introduction

Building on the advancements of large language models (LLMs), recent research has explored the development of large reasoning models (LRMs), e.g., DeepSeek-R1 (Guo et al., 2025), GPT-O1 (OpenAI, 2025), and Qwen3 (Yang et al., 2025a), thereby unlocking the potential for more complex reasoning tasks, such as mathematical reasoning (Cobbe et al., 2021) and other specialized domains (Chen et al., 2021; Phan et al., 2025). These LRMs integrate test-time compute scaling (Snell et al., 2024) and reinforcement learning techniques, enabling multi-round reasoning before arriving at a solution. By reviewing and adjusting intermediate reasoning steps during the inference process, the models enhance the reliability and correctness of the answer.

Despite the widespread use of LRMs due to their strong problem-solving capabilities for complex tasks, they often produce excessively redundant and unnecessary reasoning chains, a phenomenon commonly referred to as "overthinking" (Chen et al., 2025; Team et al., 2025). This phenomenon substantially increases computational load and inference latency. Moreover, these redundant chains may also reduce accuracy, as the LRM deviates from the correct reasoning path, resulting in repetitive and incorrect reasoning steps that hinder the overall performance (Yang et al., 2025b; Huang et al., 2026). To address "overthinking" in LRMs, current approaches have pursued three directions: training-based, prompt-based, and output-based methods. However, training-based methods incur substantial training costs, prompt-based methods struggle with task generalization, and output-based methods, although plug-and-play, rely solely on internal model confidence signals.

Existing output-based methods assume that higher model prediction confidence correlates with answer accuracy (Yang et al., 2025b). As illustrated in Figure 1, these methods typically follow a "reasoning-probing-exit" paradigm, where an early exit is triggered when the confidence at the probing stage exceeds a predefined threshold. However, these methods face significant limitations due to inherent challenges in LRMs: they often exhibit overconfidence on challenging problems (Phan et al., 2025) and insufficient confidence on easy ones (Huang et al., 2024), resulting in incorrect termination decisions and degraded response quality. We argue that the latent attentional dynamics of LRMs provide a more principled metric for evaluating internal reasoning states. This is inspired by recent advances in key-value cache eviction, where attention matrices serve as an information filter that selectively preserves tokens with high importance (Li et al., 2024; Kim et al., 2026). From an information-theoretic perspective, the resulting at-

---

[1]University of Electronic Science and Technology of China, Chengdu, China [2]Ubiquitous Intelligence and Trusted Services Key Laboratory of Sichuan Province, Chengdu, China. Correspondence to: Shuang Liang <shuangliang@uestc.edu.cn>.

*Proceedings of the 43rd International Conference on Machine Learning*, Seoul, South Korea. PMLR 306, 2026. Copyright 2026 by the author(s).

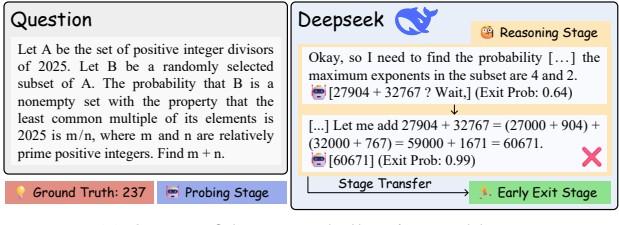

(a) Overconfidence on challenging problems.

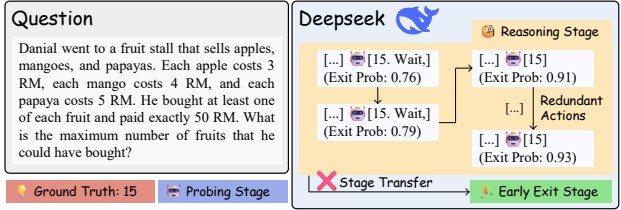

(b) Insufficient confidence on easy problems.

*Figure 1.* Two challenges of existing output-based methods. When setting an exit threshold of 0.95, (a) stops thinking incorrectly and (b) falls into redundant actions.

tention distribution yields a dual signal: it reflects historical contextual dependencies while its entropy quantifies the uncertainty of the reasoning process. Integrating these internal attention metrics with external confidence allows for a multi-dimensional quantification of reasoning state, thereby facilitating more precise and adaptive generation decisions.

To this end, we propose **A**ttention-**S**tate **A**daptive **G**eneration (**ASAG**), a training-free mechanism that adaptively adjusts reasoning strategies from the perspective of information flows. Specifically, ASAG probes the model at action transition points (ATPs) and jointly computes model confidence and attention entropy to guide the LRM generation and determines when to terminate inference. Unlike prior output-based approaches, ASAG leverages attention entropy to capture the stability of information flow, thereby alleviating premature termination caused by overconfidence on challenging problems and delayed exit due to insufficient confidence on easy ones. In addition, attention patterns are used to identify thinking traps, enabling lightweight interventions to redirect the LRM when necessary.

Our method is a training-free, output-based framework that adaptively adjusts the reasoning process, achieving a joint improvement in efficiency and accuracy. Extensive experiments on LRMs of varying parameter scales from the DeepSeek-R1-Distill and Qwen3 series show consistent performance gains across multiple reasoning benchmarks. Across most evaluations, ASAG delivers higher accuracy while maintaining comparable or improved token efficiency compared to existing state-of-the-art methods. Specifically, on Qwen3-4B and Qwen3-8B, ASAG achieves absolute accuracy improvements of 2.9% and 3.2%, respectively, while reducing the number of generated tokens by nearly 37% and 40% across all benchmarks compared to vanilla LRMs.

## 2. Related Work

Benefiting from the test-time scaling methodologies, LRMs can leverage chain-of-thought (CoT) (Wei et al., 2022) reasoning to generate explicit, step-by-step reasoning sequences before arriving at a final answer. However, many LRMs tend to produce redundant reasoning steps, making them unable to provide answers within a specific token budget. In worse cases, excessive reasoning steps introduce errors or obscure logical clarity. To mitigate the overthinking problem, existing approaches can be classified into three categories: training-based methods, prompt-based methods and output-based methods.

**Training-based methods** use supervised fine-tuning (SFT) with variable-length CoT data or designing reward metrics related to reasoning length for reinforcement learning (RL). (1) SFT methods. C3oT (Kang et al., 2025) constructs a CoT compressor that reduces original CoTs into more concise formulations and trains the model to learn the relationships between CoTs of different lengths, enabling the model to generate shorter CoTs during inference. Yu et al. (Yu et al., 2024) transfer high-quality reasoning strategies into standard LLMs through self-supervised distillation, improving output quality while reducing inference cost and latency. VeriThinker (Chen et al., 2026b) trains the model to verify the correctness of the CoT process, mitigating the "overthinking phenomenon." (2) RL methods. Yang et al. (Yang et al., 2025c) provide a comprehensive analysis of the mechanism underlying long CoT reasoning. DAST (Shen et al., 2025) employs budget-based reward shaping together with budget preference optimization, enabling LRMs to dynamically adjust the length of their CoT according to the problem difficulty. RSD (Liao et al., 2025) pairs a lightweight draft model with a target model and introduces a process reward model to dynamically decide when to invoke the target model, thereby substantially reducing computational cost.

**Prompt-based methods** aim to reduce reasoning length by designing well-crafted prompts. CoD (Xu et al., 2025) designs prompts that encourage the model to produce only the most essential and concise reasoning segments. TALE (Han et al., 2025) dynamically adjusts the reasoning token budget according to problem difficulty, thereby reducing redundant reasoning outputs and overall computational cost.

**Output-based methods** determine whether to stop reasoning early based on the LRM's intermediate outputs or hidden states. Sun et al. (Sun et al., 2024) propose Speculative Rejection, a method that employs best-of-n decoding and dynamically filters candidate responses that are unlikely to achieve high scores during generation. Fu et al. (Fu et al., 2024) introduce Certaindex to measure the stability and certainty of a model's reasoning process: when the intermediate outputs remain consistent across several reasoning steps, it indicates that the model is confident of the answer

and requires no further thinking tokens. DEER (Yang et al., 2025b) evaluates the confidence of intermediate answers at action transition points (ATP) to determine whether early stopping can be applied.

## 3. Motivations and Observations

### 3.1. Preliminaries

**Generation patterns:** Unlike traditional LLMs that directly generate an answer from the input, LRMs separate the generation process into two phases: slow thinking and conclusion. During the slow thinking stage, LRMs produce step-by-step reasoning traces, and then use thinking delimiters (e.g., $</think>$) to transition into the conclusion stage to generate the final answer. Recent research has shown that within the slow thinking phase, LRMs perform multiple reasoning actions, and the transition between these actions is indicated by action transition points (ATPs), marked by tokens such as "wait", "hmm" (Yang et al., 2025b; Luo et al., 2025). By evaluating the model's confidence in intermediate answers at these ATPs, one can determine whether early stopping is feasible, thereby improving reasoning efficiency.

**Query-key attention matrix:** Using the query-key attention matrix between the current decoding window and all global tokens, we define the normalized Shannon entropy as a metric for the dispersion of the internal attention information flow:

$$A_{h,l}^W = \text{Softmax}(A_{h,l}^S, \dim = -1), \qquad (1)$$

$$H_{h,l} = -\frac{\sum_{i=1}^{q} \sum_{j=1}^{k} A_{h,l}^W[i,j] \log A_{h,l}^W[i,j]}{\log k}, \qquad (2)$$

where $A_{h,l}^S, A_{h,l}^W, H_{h,l}$ denote the attention score matrix, the weight matrix and the normalized entropy of the $h$-th head in the $l$-th layer, respectively; $q, k$ denote the length of the query and key matrix. From a probabilistic perspective, $A_{h,l}^W[i,j]$ quantifies the influence of the $j$-th token on the $i$-th token. We hypothesize that as the LRM converges toward a reliable conclusion, the entropy will exhibit a decrease. This phenomenon reflects a transition from diffuse exploratory attention to concentrated evidence-driven attention, where the model focuses more on a small number of critical tokens.

**Experimental settings:** To validate our hypothesis above, we investigate the model's reasoning state by examining the entropy variation during reasoning. To this end, we employ Qwen3-8B (Yang et al., 2025a) in thinking mode. With an appropriate parameter scale, Qwen3-8B is among the most capable reasoning models available. We use a decoding temperature of $T$=0 and randomly sample 500 instances from the DAPO-MATH-17K dataset (Yu et al., 2026). We designate "Wait" as the ATP and probe intermediate answers by appending a prompt *"\n\n Final Answer\n\n \boxed"* to

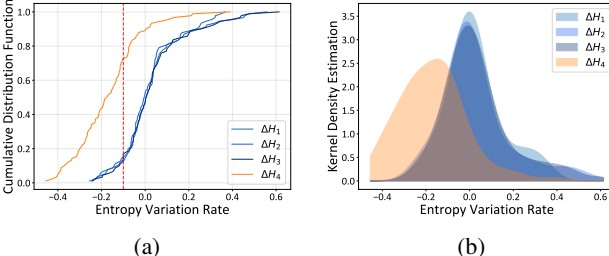

*Figure 2.* (a) The cumulative distribution function and (b) the kernel density estimation of the entropy variation rate $\Delta H$. $\Delta H_1, \Delta H_2, \Delta H_3, \Delta H_4$ denote the entropy variation rates between $H_1$ and $H_2$, $H_1$ and $H_3$, $H_1$ and $H_4$, $H_1$ and $H_f$.

the current generation. For these 500 samples, we record the entropy variation rate and the correctness of the intermediate answer at each state transition, computed as follows:

$$H^{q_1} = \sum_{h=1}^{N} \sum_{l=l'}^{L} H_{h,l}^{q_1}, \quad H^{q_2} = \sum_{h=1}^{N} \sum_{l=l'}^{L} H_{h,l}^{q_2}, \qquad (3)$$

$$\Delta H = \frac{H^{q_2} - H^{q_1}}{H^{q_1}}, \qquad (4)$$

where $N, L$ are the number of heads and layers in the LRM; $q_1, q_2$ denote the token positions of the final decoding window at the previous and current monitoring steps, respectively; $l'$ equals $L - 3$. We sum the entropy values across all attention heads in the last four model layers to obtain $H^{q_1}$ and $H^{q_2}$, and then compute the entropy variation rate to evaluate the model's reasoning state. More experimental details and explanations are provided in Appendix A.

### 3.2. Entropy Analysis

We select 274 samples with correct final answers. For each sample, we identify the state transition at which the LRM first derives a correct intermediate answer, and record the corresponding number of thinking tokens. The attention entropy at this transition is denoted as $H_f$. We then compute entropy values at four earlier stages, where the number of thinking tokens reaches 20%, 40%, 60%, and 80% of this amount, denoted as $H_1$-$H_4$, respectively. All entropy values are computed using Equations (2) and (3). Finally, following Equation (4), we compute the entropy variation rate $\Delta H$ between $H_1$ and each subsequent entropy value.

Figure 2 presents the cumulative distribution function (CDF) and the kernel density estimation (KDE) of the entropy variation rate $\Delta H$. The results show that when the model has not produced a correct answer, the attention entropy remains relatively high and stable. In contrast, once a correct answer is reached, it drops sharply. Notably, over 70% of $\Delta H_4$ samples fall below $-0.1$, whereas only a minority of samples for the other variations are less than this threshold. This divergence suggests that monitoring internal entropy dynamics provides a robust signal for adaptive early termination.

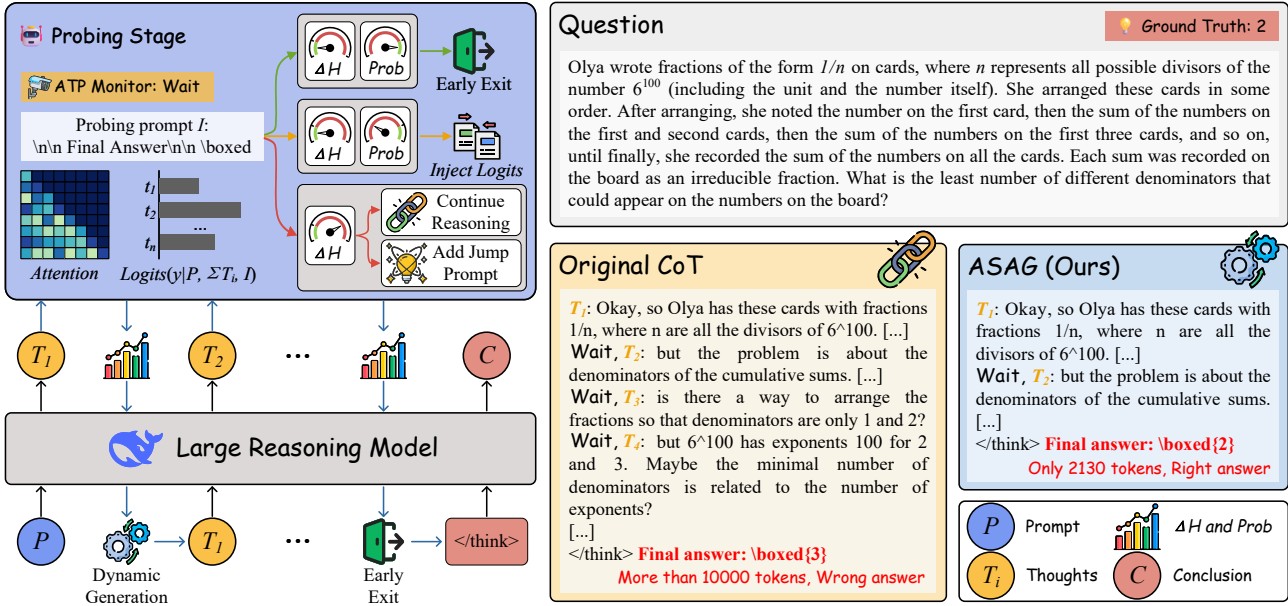

**Figure 3.** Overview of ASAG. ASAG adaptively determines subsequent generation strategies based on both attention dynamics and confidence signals, including attention-guided early exit, convergence-boosting logits injection, trap-escaping jump prompt intervention and vanilla-like generation.

## 4. Methodology

In this section, we present **A**ttention-**S**tate **A**daptive **G**eneration (ASAG), a training-free framework that dynamically guides the LRM during generation and determines when to terminate the slow thinking phase. Our approach is based on the observation that when a model converges on a reliable intermediate answer, its internal attention distribution becomes more concentrated, leading to a significant reduction in attention entropy. By integrating both model confidence and latent attention dynamics, ASAG provides a more precise assessment of the model's reasoning state beyond traditional confidence signals.

As illustrated in Figure 3, ASAG operates through three stages: reasoning, probing, and early exit. The method iteratively probes the model at natural action boundaries, i.e., ATPs, and adaptively adjusts the subsequent generation policy based on the probing outcomes. Specifically, ASAG dynamically selects from four generation strategies: attention-guided early exit, convergence-boosting logits injection, trap-escaping jump prompt intervention, and vanilla-like generation when no intervention is needed. Detailed pseudocode for ASAG is provided in Algorithm 1 of Appendix B.

### 4.1. Attention-Guided Early Exit Criterion

Given a prompt $P$, the model performs standard CoT-style generation until an ATP is detected (e.g., when the model encounters a "wait" token that marks the end of a thought

action). At each ATP, the model enters the probing stage: we append a probing prompt "*\n\n Final Answer\n\n \boxed*" to the current generation to detect the intermediate answer. Formally, the intermediate answer $A = [a_1, a_2, \ldots, a_n]$ at this ATP is obtained as: $A = \text{LRM}(P, T, I)$, where $T$ represents the generated thoughts, $I$ denotes the probing prompt. The model confidence $\mathcal{C}$ is subsequently derived from the token probabilities of $A$:

$$p(a_t) = \text{Softmax}(\mathcal{M}(P, T, I, a_{<t})), \quad \mathcal{C} = \frac{1}{n}\sum_{i=1}^{n} p(a_i),$$
(5)

where $\mathcal{M}$ is the language model head at the final layer of the LRM and $p(a_t)$ denotes the model probability of selecting token $a_t$. The model confidence $\mathcal{C}$ is computed as the average of the token probabilities over all tokens in $A$.

Then, to quantify the concentration of information flow, we compute the attention entropy across all heads in the last four layers of the LRM. Let $H_{h,l}$ be the entropy for head $h$ at layer $l$, as defined in Equation (2). We aggregate attention entropy values as:

$$H = \sum_{h=1}^{N}\sum_{l=l'}^{L} H_{h,l},$$
(6)

where $N$ and $L$ denote the number of heads and layers, respectively; $l'$ equals $L - 3$. For attention calculation, we define a decoding window to include all tokens from the probing prompt $I$ together with the same number of the most

recent tokens from the generated thoughts $T$. This ensures a comprehensive view of the model's current reasoning state.

While DEER (Yang et al., 2025b) triggers an early exit only when the model confidence $\mathcal{C}$ exceeds an empirical threshold $\lambda$, ASAG refines this criterion by incorporating entropy variation. Let $H_1$ and $H$ represent the entropy values at the first ATP and a subsequent ATP, respectively, as calculated by Equation (6). We define the entropy variation rate as $\Delta H = \frac{H - H_1}{H_1}$. Our early exit decision rules are as follows: at the first ATP, the LRM exits if $\mathcal{C} > \lambda$. At any subsequent ATP, an early exit is permitted only if $\mathcal{C} > \lambda$ and $\Delta H < \alpha$; otherwise, the reasoning process is deemed unstable, and the LRM continues generation. By integrating model confidence with attention entropy, ASAG mitigates the risk of premature termination caused by model overconfidence on challenging tasks.

### 4.2. Convergence-Boosting Logits Injection Strategy

Furthermore, in cases where reasoning has converged despite low model confidence, i.e., $\mathcal{C} < \lambda$ and $\Delta H < \alpha$, we infer that the LRM has successfully captured the relevant evidence though the token-level confidence is low. While existing methods continue to follow the vanilla LRM reasoning trajectory until the confidence threshold is reached, which often results in overthinking and hesitation, as shown in Figure 1(b). In contrast, we aim to leverage the converged intermediate answers to guide subsequent generation, thereby shifting the conventional reasoning paradigm. Although one could directly modulate attention patterns using the extracted key information (Li et al., 2026b; Liu et al., 2024), such approaches often incur substantial computational overhead. To this end, we propose a lightweight and easily integrated strategy that modifies the output logits, necessitating only minimal adjustments to the language model head $\mathcal{M}$. Specifically, to mitigate insufficient confidence on easy problems, we extract the normalized logits probability of the intermediate answer $a_r$ in $A$, denoted as $Logits_r$, and inject it to guide the generation process. The adjusted logits for the subsequent tokens are defined as follows:

$$Logits = 0.95 \cdot \text{Softmax}(\mathcal{M}(P, T)) + 0.05 \cdot Logits_r. \quad (7)$$

This lightweight logits injection guides the LRM toward the converged intermediate answer, facilitating earlier commitment to correct conclusions.

### 4.3. Trap-Escaping Jump Prompt Intervention

Lastly, a large entropy variation rate $\Delta H$ at an ATP indicates that the LRM's reasoning process has not yet converged, and further generation is required. However, Ding et al. (Ding et al., 2025) point out that LRMs can become stuck in a thinking trap, where an incorrect initial reasoning path leads LRMs to persist in an unproductive CoT, ultimately failing

to reach the correct solution. In such instances, simply continuing generation is ineffective; instead, the LRM must be redirected toward an alternative reasoning trajectory.

Drawing inspiration from SBT (Zhao et al., 2026), which suggests that explicitly prompting LRMs with breaking prompts can help interrupt reasoning and encourage early termination, we design a mechanism to identify these thinking traps and apply jump prompts $J$ to encourage the LRM to restart reasoning from a different perspective. Specifically, at each ATP, we construct a global attention weight matrix $A_{global}^W$ that characterizes the attention distribution of the current decoding window across all tokens:

$$A_{global}^W = \frac{1}{N} \cdot \frac{1}{4} \sum_{h=1}^{N} \sum_{l=L-3}^{L} A_{h,l}^W. \quad (8)$$

If the average token-level attention allocated to the previous thought action $T_{i-1}$ exceeds that of the current thought action $T_i$, we regard this as a sign of a thinking trap, where the model repetitively focuses on prior reasoning without making substantive progress in reasoning. In such cases, we add a jump prompt $J$ *"Wait, my previous reasoning is not correct. I should adopt a more concise and different approach to reexamine this problem.\n\n"* to the current generation to encourage a transition toward a different reasoning trajectory. Our empirical analysis suggests that jump prompts do not always induce a new perspective. This limitation may arise from deeply ingrained reasoning biases or persistent insufficient confidence. To prevent meaningless computation, we establish a maximum threshold $s$ for jump attempts; once this limit is exceeded, the system terminates reasoning and triggers an early exit.

In summary, ASAG evaluates the model's reasoning state using both model confidence and attention entropy. This combination mitigates both overconfidence on challenging problems and insufficient confidence on easy ones.

## 5. Experiments

### 5.1. Experimental Setup

**Benchmarks.** We evaluate our method on nine classic reasoning benchmarks, including six mathematical reasoning benchmarks: GSM8K (Cobbe et al., 2021), MATH-500 (Hendrycks et al., 2021), AMC 2023 (AI-MO, 2024), AIME 2024 (MAA Committees), AIME 2025 (MAA Committees) and OlympiadBench (He et al., 2024), one scientific reasoning benchmark: GPQA Diamond (Rein et al., 2023), and two code reasoning benchmarks: HumanEval (Chen et al., 2021), LiveCodeBench (Jain et al., 2025). More benchmark details are provided in Appendix D.1.

**Metrics.** To comprehensively evaluate the performance, we

*Table 1.* Comparison across LRMs of different scales and multiple baselines. The best and the second best results are highlighted in **bold** and underline, respectively.

| Method | GSM8K | | | MATH-500 | | | AIME 2024 | | | OlympiadBench | | | GPQA Diamond | | | AVG | |
|---|---|---|---|---|---|---|---|---|---|---|---|---|---|---|---|---|---|
| | Acc↑ | Len↓ | CR↓ | Acc↑ | Len↓ | CR↓ | Acc↑ | Len↓ | CR↓ | Acc↑ | Len↓ | CR↓ | Acc↑ | Len↓ | CR↓ | Acc↑ | CR↓ |
| | | | | | | | *Qwen3-4B* | | | | | | | | | | |
| Vanilla | 93.8 | 2,142 | 100% | 92.4 | 4,910 | 100% | 63.3 | 11,916 | 100% | 59.0 | 8,915 | 100% | 46.5 | 9,277 | 100% | 71.0 | 100% |
| NoThinking | 89.6 | 624 | 29.1% | 84.2 | 1,013 | 20.6% | 23.3 | 6,458 | 54.2% | 40.6 | 4,358 | 48.9% | 36.4 | 1,738 | 18.7% | 54.8 | **34.3%** |
| TALE | 91.3 | 926 | 43.2% | 87.6 | 2,827 | 57.6% | 60.0 | 11,469 | 96.2% | 54.7 | 6,068 | 68.1% | 41.9 | 2,620 | 28.2% | 67.1 | 58.7% |
| Dynasor | 92.9 | 938 | 43.8% | 91.6 | 2,965 | 60.4% | 63.3 | 11,387 | 95.6% | 63.6 | 6,863 | 77.0% | 46.5 | 4,179 | 45.0% | 71.6 | 64.4% |
| DEER | 94.2 | 1,286 | 60.0% | 92.6 | 3,197 | 65.1% | 60.0 | 8,745 | 73.4% | 62.9 | 7,344 | 82.4% | 47.0 | 3,802 | 41.0% | 71.3 | 64.4% |
| ASAG (**ours**) | 94.2 | 1,139 | 53.2% | 92.8 | 3,325 | 67.7% | 70.0 | 8,768 | 73.6% | 64.6 | 7,536 | 84.5% | 48.0 | 3,563 | 38.4% | **73.9** | 63.5% |
| | | | | | | | *Qwen3-8B* | | | | | | | | | | |
| Vanilla | 94.6 | 2,338 | 100% | 92.2 | 4,926 | 100% | 63.3 | 12,101 | 100% | 59.9 | 9,268 | 100% | 52.5 | 9,382 | 100% | 72.5 | 100% |
| NoThinking | 91.3 | 685 | 29.3% | 87.4 | 1,480 | 30.0% | 23.3 | 7,121 | 58.8% | 48.7 | 5,219 | 56.3% | 48.5 | 2,564 | 27.3% | 59.8 | **40.3%** |
| TALE | 92.5 | 1,142 | 48.8% | 91.8 | 3,682 | 74.7% | 60.0 | 11,847 | 97.9% | 56.1 | 7,306 | 78.8% | 52.0 | 4,161 | 44.4% | 70.5 | 68.9% |
| Dynasor | 94.2 | 1,024 | 43.8% | 92.4 | 3,361 | 68.2% | 60.0 | 11,365 | 93.9% | 62.4 | 7,640 | 82.4% | 55.1 | 5,647 | 60.2% | 72.8 | 69.7% |
| DEER | 95.0 | 1,096 | 46.9% | 92.4 | 2,966 | 60.2% | 63.3 | 8,930 | 73.8% | 61.7 | 7,367 | 79.5% | 54.5 | 5,334 | 56.9% | 73.4 | 63.5% |
| ASAG (**ours**) | 95.9 | 1,063 | 45.5% | 93.0 | 3,152 | 64.0% | 66.7 | 8,683 | 71.7% | 63.9 | 7,296 | 78.7% | 56.1 | 5,714 | 60.9% | **75.1** | 64.2% |
| | | | | | | | *Qwen3-14B* | | | | | | | | | | |
| Vanilla | 95.1 | 2,082 | 100% | 93.4 | 4,725 | 100% | 70.0 | 10,537 | 100% | 61.9 | 8,519 | 100% | 58.6 | 7,688 | 100% | 75.8 | 100% |
| NoThinking | 93.5 | 447 | 21.5% | 88.0 | 1,403 | 29.7% | 30.0 | 7,726 | 73.3% | 50.4 | 5,462 | 64.1% | 50.5 | 2,308 | 30.0% | 62.5 | **43.7%** |
| TALE | 94.7 | 963 | 46.3% | 93.2 | 3,616 | 76.5% | 70.0 | 10,358 | 98.3% | 58.4 | 7,250 | 85.1% | 59.1 | 5,391 | 70.1% | 75.1 | 75.3% |
| Dynasor | 95.6 | 1,364 | 65.5% | 93.8 | 3,958 | 83.8% | 70.0 | 11,156 | 105.9% | 63.0 | 7,336 | 86.1% | 59.6 | 5,387 | 70.1% | 76.4 | 82.3% |
| DEER | 95.3 | 863 | 41.5% | 94.0 | 3,410 | 72.2% | 73.3 | 8,402 | 79.7% | 62.7 | 7,020 | 82.4% | 60.6 | 4,998 | 65.0% | 77.2 | 68.2% |
| ASAG (**ours**) | 96.2 | 1,039 | 49.9% | 95.0 | 3,505 | 74.2% | 73.3 | 7,602 | 72.1% | 64.6 | 6,819 | 80.0% | 63.1 | 5,075 | 66.0% | **78.4** | 68.4% |
| | | | | | | | *DeepSeek-R1-Distill-Qwen-7B* | | | | | | | | | | |
| Vanilla | 89.5 | 1,452 | 100% | 87.8 | 3,790 | 100% | 40.0 | 13,722 | 100% | 48.6 | 8,347 | 100% | 25.3 | 9,834 | 100% | 58.2 | 100% |
| NoThinking | 86.3 | 328 | 22.6% | 80.0 | 1,167 | 30.8% | 16.7 | 7,365 | 53.7% | 37.3 | 3,140 | 37.6% | 26.8 | 1,385 | 14.1% | 49.4 | **31.8%** |
| TALE | 88.2 | 909 | 62.6% | 88.6 | 2,738 | 72.2% | 33.3 | 11,627 | 84.7% | 48.9 | 5,711 | 68.4% | 26.3 | 4,883 | 49.7% | 57.1 | 67.5% |
| Dynasor | 89.7 | 1,258 | 86.6% | 89.4 | 2,994 | 79.0% | 40.0 | 11,462 | 83.5% | 50.0 | 5,973 | 71.6% | 29.3 | 6,817 | 69.3% | 59.7 | 78.0% |
| DEER | 90.3 | 967 | 66.6% | 89.2 | 2,407 | 63.5% | 43.3 | 8,669 | 63.2% | 51.1 | 5,420 | 64.9% | 28.8 | 5,639 | 57.3% | 60.5 | 63.1% |
| ASAG (**ours**) | 91.0 | 1,009 | 69.5% | 89.8 | 2,533 | 66.8% | 46.7 | 9,154 | 66.7% | 52.3 | 5,325 | 63.8% | 31.8 | 5,584 | 56.8% | **62.3** | 64.7% |

adopt three metrics. (1) pass@1 accuracy (Acc) measures the fraction of problems that the model solves correctly in a single generation. (2) We compute the average number of generated tokens (Len) to evaluate the inference-time reasoning cost. (3) Compression rate (CR) is defined as the ratio of the average response length to that of the vanilla baseline. Higher Acc, lower Len, and lower CR indicate better performance.

**Backbone LRMs.** Our evaluation covers various open-source models with varying parameter scales. Specifically, we consider the DeepSeek-R1-Distill series (including Qwen-7B and Llama-8B) (Guo et al., 2025), and the Qwen3 series (including 4B, 8B, 14B, and 32B) (Yang et al., 2025a).

**Baselines.** We evaluate ASAG against three categories of baselines: (i) Vanilla: standard decoding without any intervention; (ii) Prompt-based methods: including NoThinking (Ma et al., 2025) that skips intermediate reasoning for direct answer generation, and TALE (Han et al., 2025) that constrains reasoning within a predefined token budget; and (iii) output-based methods, including Dynasor (Fu et al., 2024) that periodically probes intermediate answers at fixed token intervals and terminates generation when multiple consecutive answers are consistent, and DEER (Yang et al., 2025b) that dynamically truncates generation by detecting high-confidence intermediate answers at ATPs such as "Wait".

**Implementations.** All models are deployed with the "bfloat16" data format due to the balance between efficiency and performance. We set the generation mode to greedy-decoding for all methods with deterministic decoding parameters: do_sample=False, temperature=0, top_p=1, max_new_tokens=16000. We set the hyperparameters $\lambda$ to 0.95, $\alpha$ to $-0.1$, jump attempts $s$ to 1.

### 5.2. Main Results

Table 1 shows the main experimental results on four representative mathematical reasoning benchmarks and one scientific reasoning benchmark across four reasoning models. We also provide further results across other benchmarks covering DeepSeek-R1-Distill-Llama-8B and Qwen3-32B in Appendix E. These results reveal four principal insights:

(i) Compared to the vanilla baseline, ASAG consistently improves accuracy while significantly reducing the length of generated tokens across most benchmarks and LRMs, without additional training. These results demonstrate the robustness and generalization of our approach, suggesting that ASAG is agnostic to both model architectures and datasets. Specifically, on the five benchmarks reported in Table 1, ASAG yields the largest average performance gain on DeepSeek-R1-Distill-Qwen-7B, delivering a 4.1% accuracy improvement with a compression ratio of 64.7%.

*Table 2.* Ablation study of ASAG. w/o $Logits_r$ denotes removing the logits injection strategy in Equation (7).

| Method | GSM8K | | | MATH-500 | | | AIME 2024 | | | OlympiadBench | | | GPQA Diamond | | | AVG | |
|---|---|---|---|---|---|---|---|---|---|---|---|---|---|---|---|---|---|
| | Acc↑ | Len↓ | CR↓ | Acc↑ | Len↓ | CR↓ | Acc↑ | Len↓ | CR↓ | Acc↑ | Len↓ | CR↓ | Acc↑ | Len↓ | CR↓ | Acc↑ | CR↓ |
| *Qwen3-4B* | | | | | | | | | | | | | | | | | |
| ASAG | 94.2 | 1,139 | 53.2% | 92.8 | 3,325 | 67.7% | 70.0 | 8,768 | 73.6% | 64.6 | 7,536 | 84.5% | 48.0 | 3,563 | 38.4% | **73.9** | **63.5%** |
| w/o $Logits_r$ | 94.0 | 1,206 | 56.3% | 92.8 | 3,483 | 70.9% | 67.7 | 9,463 | 79.4% | 64.2 | 7,827 | 87.8% | 48.5 | 3,760 | 40.5% | 73.4 | 67.0% |
| *Qwen3-8B* | | | | | | | | | | | | | | | | | |
| ASAG | 95.9 | 1,063 | 45.5% | 93.0 | 3,152 | 64.0% | 66.7 | 8,683 | 71.7% | 63.9 | 7,296 | 78.7% | 56.1 | 5,714 | 60.9% | **75.1** | **64.2%** |
| w/o $Logits_r$ | 95.9 | 1,185 | 50.7% | 93.4 | 3,570 | 72.5% | 66.7 | 9,170 | 75.8% | 63.6 | 7,852 | 84.7% | 56.1 | 6,138 | 65.4% | **75.1** | 69.8% |
| *Qwen3-14B* | | | | | | | | | | | | | | | | | |
| ASAG | 96.2 | 1,039 | 49.9% | 95.0 | 3,505 | 74.2% | 73.3 | 7,602 | 72.1% | 64.6 | 6,819 | 80.0% | 63.1 | 5,075 | 66.0% | **78.4** | **68.4%** |
| w/o $Logits_r$ | 96.1 | 1,074 | 51.6% | 94.4 | 3,772 | 79.8% | 73.3 | 8,774 | 83.3% | 64.3 | 7,144 | 83.9% | 63.1 | 5,099 | 66.3% | 78.2 | 72.3% |
| *DeepSeek-R1-Distill-Qwen-7B* | | | | | | | | | | | | | | | | | |
| ASAG | 91.0 | 1,009 | 69.5% | 89.8 | 2,533 | 66.8% | 46.7 | 9,154 | 66.7% | 52.3 | 5,325 | 63.8% | 31.8 | 5,584 | 56.8% | 62.3 | **64.7%** |
| w/o $Logits_r$ | 91.2 | 1,275 | 87.8% | 89.6 | 2,669 | 70.4% | 46.7 | 9,859 | 71.8% | 52.1 | 5,429 | 65.0% | 32.3 | 6,145 | 62.5% | **62.4** | 71.5% |

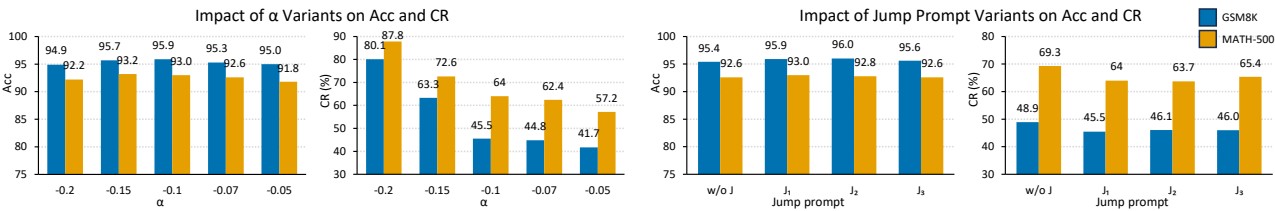

*Figure 4.* Impact of $\alpha$ variants (left) and jump prompt variants (right) on Acc and CR. Jump prompts $J_1, J_2, J_3$ are defined in Appendix D.2, and additional results for Qwen3-14B are shown in Figure 7 in Appendix F.

(ii) Compared to prompt-based methods, although NoThinking achieves extreme compression by skipping the reasoning stage, it suffers from degraded performance, particularly on challenging benchmarks such as AIME 2024, Olympiad-Bench and GPQA Diamond. TALE consistently achieves accuracy comparable to the vanilla baseline with modest compression ratios. For the output-based methods, Dynasor triggers early exit based on answer consistency; however, this often leads to premature stopping and incomplete reasoning. DEER relies on the model's confidence signals and achieves the second-best performance on most benchmarks. Our method goes beyond these approaches by jointly considering model confidence and attention entropy, enabling more precise, adaptive, and robust guidance of the generation process.

(iii) On relatively simple benchmarks such as GSM8K and MATH-500, our method achieves remarkable compression ratios. Specifically, on Qwen3-8B, ASAG attains compression ratios of 45.5% and 64.0% on these datasets, respectively, while maintaining extremely high accuracy. These results indicate that by injecting the logits from converged reasoning states, ASAG enables the LRM to commit to reliable answers earlier, mitigating insufficient confidence on easy problems. On more challenging benchmarks such as AIME 2024, OlympiadBench, and GPQA Diamond, the compression gains are less pronounced. Nevertheless, ASAG consistently yields the highest accuracy across these benchmarks. By jointly leveraging model confidence and attention entropy, we effectively alleviate the tendency of LRMs to exhibit overconfidence on challenging problems.

(iv) As shown in Table 7 in Appendix E, ASAG consistently surpasses the vanilla baseline on code reasoning benchmarks, demonstrating strong versatility. We attribute this improvement to the observation that, in code reasoning, the critical solution often hinges on few key lines of code. Once the LRM generates these essential components, further reasoning becomes unnecessary, allowing early termination without compromising correctness.

### 5.3. Ablation Study

In this section, we conduct ablation studies to evaluate the effectiveness of each component in ASAG. Since the confidence threshold $\lambda$ has been thoroughly analyzed in DEER (Yang et al., 2025b), we mainly focus our analysis on the novel components related to attention entropy.

**Analysis of entropy threshold $\alpha$.** Building on the empirical observations in Section 3, which associate reasoning convergence with a reduction in attention entropy, we introduce a threshold $\alpha$ to determine whether the reasoning process has converged. A smaller $\alpha$ enforces a stricter convergence criterion, requiring a more concentrated information flow before triggering an exit. As shown in Figures 4 and 7 (left), when $\alpha$ is set to smaller values, e.g., $-0.2$, compression gains decrease significantly, as only a few samples exhibit such pronounced entropy drops, causing ASAG to degen-

*Table 3.* Performance under the sampling setting. We conduct three sampling runs per instance and report the averaged results to ensure stability and reliability.

| Method | GSM8K | | | MATH-500 | | | AIME 2024 | | | OlympiadBench | | | GPQA Diamond | | | AVG | |
|---|---|---|---|---|---|---|---|---|---|---|---|---|---|---|---|---|---|
| | Acc↑ | Len↓ | CR↓ | Acc↑ | Len↓ | CR↓ | Acc↑ | Len↓ | CR↓ | Acc↑ | Len↓ | CR↓ | Acc↑ | Len↓ | CR↓ | Acc↑ | CR↓ |
| *Qwen3-4B* | | | | | | | | | | | | | | | | | |
| Vanilla | 94.1 | 2,307 | 100% | 92.6 | 5,082 | 100% | 64.4 | 11,639 | 100% | 59.8 | 9,172 | 100% | 46.5 | 9,130 | 100% | 71.5 | 100% |
| ASAG | 94.0 | 1,158 | 50.2% | 92.9 | 3,315 | 65.2% | 70.0 | 8,861 | 76.1% | 64.8 | 7,205 | 78.6% | 48.1 | 3,728 | 40.8% | **74.0** | **62.2%** |
| *Qwen3-8B* | | | | | | | | | | | | | | | | | |
| Vanilla | 95.6 | 2,276 | 100% | 92.7 | 5,250 | 100% | 63.3 | 12,383 | 100% | 60.4 | 9,127 | 100% | 52.9 | 9,344 | 100% | 73.0 | 100% |
| ASAG | 95.8 | 944 | 41.5% | 93.1 | 3,184 | 60.6% | 66.7 | 8,550 | 69.0% | 64.3 | 7,443 | 81.5% | 55.9 | 5,893 | 63.1% | **75.2** | **63.1%** |
| *Qwen3-14B* | | | | | | | | | | | | | | | | | |
| Vanilla | 95.6 | 1,970 | 100% | 93.9 | 4,838 | 100% | 72.2 | 10,739 | 100% | 62.3 | 8,691 | 100% | 59.8 | 7,832 | 100% | 76.8 | 100% |
| ASAG | 96.3 | 1,063 | 54.0% | 94.8 | 3,554 | 73.5% | 72.2 | 7,874 | 73.3% | 65.2 | 6,781 | 78.0% | 62.9 | 5,226 | 66.7% | **78.3** | **69.1%** |
| *DeepSeek-R1-Distill-Qwen-7B* | | | | | | | | | | | | | | | | | |
| Vanilla | 89.7 | 1,407 | 100% | 87.6 | 3,840 | 100% | 40.0 | 13,484 | 100% | 48.7 | 8,467 | 100% | 25.6 | 9,661 | 100% | 58.3 | 100% |
| ASAG | 91.1 | 1,073 | 76.3% | 89.6 | 2,651 | 69.0% | 45.6 | 9,292 | 68.9% | 52.6 | 5,136 | 60.7% | 31.7 | 5,469 | 56.6% | **62.1** | **66.3%** |

erate toward the vanilla baseline. Conversely, a larger $\alpha$ increases the risk of premature termination by mistaking minor local fluctuations in attention for logical convergence, yielding higher compression ratios at the expense of reasoning accuracy. Overall, our results reveal a "sweet spot" at $\alpha = -0.1$, which consistently delivers the most stable performance across models and benchmarks. Notably, this choice aligns with the empirical distribution of entropy variation identified in Section 3, where over 70% of correct reasoning steps fall below this threshold.

**Effect of injecting** $Logits_r$**.** As illustrated in Figure 3, when $\Delta H < \alpha$ and $\mathcal{C} > \lambda$, we regard the reasoning process as converged and guide subsequent generation via logits fusion. This design aims to strengthen the model's confidence in the converged answer and suppress unnecessary generation. Table 2 reports the results obtained after removing the logits injection component. While the Acc remains comparable to that of ASAG, the CR is substantially higher than ASAG. Specifically, ASAG achieves an absolute compression ratio improvement of 5.6% and 6.8% on Qwen3-8B and DeepSeek-R1-Distill-Qwen-7B, respectively. These results demonstrate that injecting logits effectively alleviates the issue of insufficient confidence on easy problems, enabling the model to reach the final answer earlier.

**Effect of adding jump prompts.** Jump prompts are designed to help the model escape thinking traps, preventing it from repeatedly following incorrect reasoning trajectories. Figures 4 and 7 (right) report the performance under different jump prompt designs. The results show that incorporating jump prompts consistently improves both accuracy and compression ratio, demonstrating their effectiveness in steering the model out of thinking traps. This effect is particularly important because initial reasoning errors often cause excessively long and redundant generations. Furthermore, all jump prompts yield noticeable gains, indicating that ASAG is robust to different jump prompt designs.

### 5.4. Results under Different Sampling Strategies

To comprehensively evaluate the robustness of our method, we extend our experiments from deterministic greedy decoding to stochastic sampling settings. While most reasoning models currently adopt greedy decoding to produce deterministic outputs, evaluating ASAG under sampling-based generation provides valuable insights into its practical applicability. Table 3 presents the performance results under a sampling configuration with top-p=0.95 and temperature of $T$=0.6 across five reasoning benchmarks. For each instance, we conduct three sampling runs and report the averaged results to ensure stability and reliability. Across all LRMs, our method consistently achieves substantial token reduction while improving accuracy. Specifically, on Qwen3-4B, ASAG reduces generated tokens by 37.8% while improving the average accuracy from 71.5% to 74.0%. On Qwen3-8B, ASAG achieves an average token reduction of 36.9% together with a 2.2% improvement in average accuracy. The consistent gains across different LRMs and benchmarks highlight the strong generalizability of our approach.

These findings confirm that the adaptive generation strategy of ASAG extends beyond greedy decoding and generalizes effectively to sampling configurations. The ability to maintain robust performance under sampling further suggests that models' internal attention entropy serves as a reliable indicator of reasoning state regardless of the underlying decoding strategy. This versatility makes ASAG suitable for diverse application scenarios that require a balance between output diversity and reasoning efficiency.

### 5.5. Further Discussions

**Efficiency analysis.** To evaluate the practical efficiency of ASAG, we measure end-to-end latency on MATH-500 and GPQA Diamond using a single NVIDIA A800 80G GPU. As shown in Figure 5, while ASAG introduces extra forward passes during probing at each Action Transition Point (ATP),

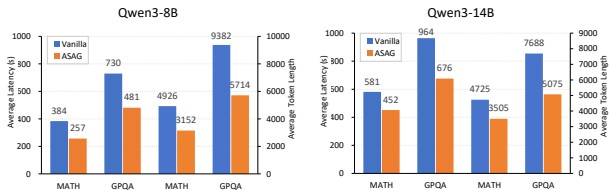

*Figure 5.* Efficiency analyses of the vanilla method and ASAG on MATH-500 and GPQA Diamond benchmarks. ASAG achieves consistent improvement on both end-to-end latency and generated token length.

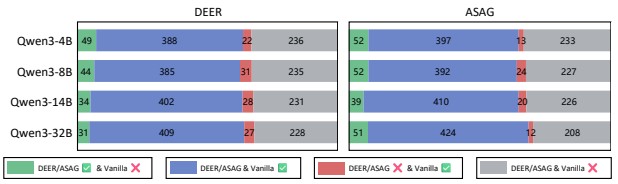

*Figure 6.* Analysis of OlympiadBench results on DEER and ASAG. ✓ and ✗ denote samples with correct and wrong answers.

*Table 4.* Results with different ATP choices. All experiments are conducted on Qwen3-8B. ASAG(W) and ASAG(A) denote ASAG with "Wait" and "Alternatively" as ATP signals, respectively. Chunk Size denotes the average number of generated tokens within a single reasoning chunk, and Exit Acc denotes the answer accuracy for early exit instances.

| Method | Acc | Len | Chunk Size | Exit Acc |
|---|---|---|---|---|
| | | ***MATH-500*** | | |
| Vanilla | 92.2 | 4,926 | - | - |
| ASAG(W) | 93.0 | 3,152 | 316.6 | 96.3% |
| ASAG(A) | 92.8 | 3,220 | 784.5 | 94.8% |
| | | ***AIME 2024*** | | |
| Vanilla | 63.3 | 12,101 | - | - |
| ASAG(W) | 66.7 | 8,683 | 473.8 | 100% |
| ASAG(A) | 66.7 | 9,012 | 952.4 | 93.3% |

this computational overhead is negligible compared to the massive reduction in generated tokens. The probing stage accounts for less than 10% of total inference time. As a result, ASAG achieves a notable end-to-end speedup of approximately $1.51\times$ on Qwen3-8B. Furthermore, the early-stop mechanism effectively limits the growth of the Key-Value cache, alleviating memory pressure in complex reasoning. Overall, ASAG improves accuracy while offering a better trade-off between reasoning depth and computational cost, making it highly suitable for latency-sensitive deployment.

**Error analysis.** We conduct an error analysis to examine ASAG's effectiveness in alleviating overconfidence and insufficient confidence. First, regarding overconfidence on challenging problems, we analyze samples from Olympiad-Bench, a high-difficulty benchmark, and adopt the state-of-the-art confidence-based method DEER (Yang et al., 2025b) as the baseline. As shown in Figure 6, the number of samples incorrectly answered by both DEER and the vanilla model is comparable to that of both ASAG and the vanilla model (grey part), suggesting that these cases are beyond the base capacity of the LRM. However, a clear performance gap appears in cases where only the vanilla model produces the correct answer (red part). In these instances, DEER often triggers a premature exit due to superficial initial reasoning and unreliable confidence signals. In contrast, ASAG recognizes high attention entropy as an unconverged reasoning state and enforces continued generation even under high confidence. This enables the LRM to complete necessary reasoning steps that confidence-based methods would truncate, thereby narrowing the gap between internal certainty and logical convergence.

**Performance with different ATP choices.**

In the main experiments, we use "Wait" as the ATP for early exit, which yields satisfactory results. To further evaluate the robustness of ASAG to ATP selection, we conduct additional experiments using "Alternatively" as the ATP. Table 4 presents the results of ASAG on MATH-500 and AIME 2024 benchmarks under these two ATP configurations on Qwen3-8B. ASAG(W) achieves accuracy improvements of 0.8% and 3.4%, together with reductions of 36.0% and 28.2% in generated tokens on MATH-500 and AIME 2024, respectively. Similarly, ASAG(A) maintains competitive performance, improving accuracy by 0.6% and 3.4% while reducing token generation by 34.6% and 25.5% on the two benchmarks. Both configurations consistently outperform vanilla generation. In addition, across benchmarks of varying difficulty, both ATP choices exhibit consistent patterns: token generation increases with task complexity, while exit accuracy remains stable. This observation further shows that ASAG can adaptively adjust its generation strategy according to reasoning complexity without relying on specific ATP choices. We attribute this robustness to two main factors. First, these markers share similar semantic functions in indicating reasoning state transitions. Second, ASAG relies on internal attention entropy, which provides a universal reasoning state measurement independent of specific linguistic markers, thereby ensuring stable performance.

## 6. Conclusion

We present Attention-State Adaptive Generation (ASAG), a training-free, plug-and-play framework that mitigates two limitations in LRMs. By leveraging latent attention dynamics, ASAG provides a principled measure of a model's reasoning state beyond traditional confidence signals. Extensive evaluations on nine benchmarks show that ASAG improves accuracy while maintaining efficiency, reducing generated tokens by 40% on Qwen3-8B. Our results demonstrate that monitoring internal information flows is crucial for building more efficient and reliable reasoning agents.

## Acknowledgments

This work is supported by the Noncommunicable Chronic Diseases-National Science and Technology Major Project No.2023ZD0501806, National Natural Science Foundation of China No.62406057, the Fundamental Research Funds for the Central Universities No.ZYGX2025XJ042, and the Sichuan Science and Technology Program under Grant No.2024ZDZX0011.

## Impact Statement

This work introduces ASAG, a method designed to enhance the efficiency and reliability of large reasoning models (LRMs). The broader impacts of ASAG are multifaceted. First, from an environmental and economic perspective, ASAG reduces token generation by nearly 40% on Qwen3-8B, directly lowering computational costs, energy consumption, and carbon footprint of large-scale AI deployments while democratizing access for resource-constrained researchers. Second, regarding AI safety and reliability, ASAG mitigates the overthinking and overconfidence issues in current LRMs through adaptive generation mechanisms. This enables more self-aware AI systems that better estimate reasoning progress, which is crucial for decision-sensitive domains like education and scientific research where reasoning accuracy and transparency are essential. We do not foresee any negative social impacts from this work, as it primarily serves as a general-purpose optimization for existing model architectures.

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

## A. Details in Section 3

In Equation (3), $q_1$ and $q_2$ denote the token positions of the decoding window at the previous and current monitoring steps, respectively. We set the window length to 32 to obtain a more reliable estimation of the model's attention dynamics. In addition, $l'$ in Equation (3) is set to $L-3$. This design is motivated by previous studies on information flow in LLMs (Wang et al., 2023; Yuan et al., 2024; Li et al., 2026b), which show that attention in shallow layers tends to be more dispersed, corresponding to information aggregation and input understanding, whereas attention in deeper layers becomes more concentrated, reflecting information extraction and the enhancement of key information for response generation. Therefore, we compute the attention entropy $H$ only over last four model layers, where reasoning convergence is characterized by focused attention rather than the dispersed patterns typically observed in earlier layers.

## B. Pseudocode of ASAG

---

**Algorithm 1** Attention-State Adaptive Generation (ASAG)

---

**Initialization:** Large Reasoning Model LRM$(\cdot)$, zero-shot-CoT $zs\_cot$, question, probing prompt $I$, jump prompt $J$, set of action transition points $\mathbb{P}$, end-of-thinking delimiter $\langle$/think$\rangle$, maximum length $max\_len$, confidence threshold $\lambda$, entropy variation threshold $\alpha$, maximum jump attempts $s$.

$\mathbf{x} \leftarrow zs\_cot +$ question, $\mathbf{r} \leftarrow []$, $H_1 \leftarrow$ None, $\Delta H \leftarrow$ None, $jump\_count \leftarrow 0$

**while** $len(\mathbf{x}) < max\_len$ **do**
  $y \leftarrow$ LRM$(\mathbf{x})$
  **if** $y \in \mathbb{P}$ **then**
    $A \leftarrow$ LRM$(\mathbf{x} + I)$
    Compute model confidence $\mathcal{C}$ and attention entropy $H$ according to Equation (5) and (6).
    **if** $H_1 =$ None **then**
      $H_1 \leftarrow H$
    **else**
      $\Delta H \leftarrow \dfrac{H - H_1}{H_1}$
    **end if**
    **if** ($\Delta H =$ None **and** $\mathcal{C} > \lambda$) **or** ($\Delta H < \alpha$ **and** $\mathcal{C} > \lambda$) **then**
      $\mathbf{x} \leftarrow \mathbf{x} + \langle$/think$\rangle$, $\mathbf{r} \leftarrow \mathbf{r} + \langle$/think$\rangle$
    **else if** $\Delta H < \alpha$ **and** $\mathcal{C} < \lambda$ **then**
      Convergence-boosting logits injection following Equation (7)
    **else if** $\Delta H > \alpha$ **then**
      Compute global attention matrix $A_{global}^{W}$ following Equation (8)
      **if** $\frac{1}{T_{i-1}} A_{global}^{W}[\ldots, T_{i-1}] > \frac{1}{T_i} A_{global}^{W}[\ldots, T_i]$ **then**
        **if** $jump\_count \leq s$ **then**
          $\mathbf{x} \leftarrow \mathbf{x} + J$, $\mathbf{r} \leftarrow \mathbf{r} + J$, $jump\_count \leftarrow jump\_count + 1$
        **else**
          $\mathbf{x} \leftarrow \mathbf{x} + \langle$/think$\rangle$, $\mathbf{r} \leftarrow \mathbf{r} + \langle$/think$\rangle$
        **end if**
      **else**
        $\mathbf{x} \leftarrow \mathbf{x} + y$, $\mathbf{r} \leftarrow \mathbf{r} + y$
      **end if**
    **end if**
  **else**
    $\mathbf{x} \leftarrow \mathbf{x} + y$, $\mathbf{r} \leftarrow \mathbf{r} + y$
  **end if**
**end while**
**return r**

---

## C. Symbol Explanation

Table 5 provides a comprehensive list of symbols used in this paper and their corresponding descriptions.

*Table 5.* Symbols and Descriptions.

| Symbol | Description |
| --- | --- |
| $A_{h,l}^S$ | the attention score matrix of the $h$-th head in the $l$-th layer |
| $A_{h,l}^W$ | the attention weight matrix of the $h$-th head in the $l$-th layer |
| $H_{h,l}$ | the normalized Shannon entropy value of the $h$-th head in the $l$-th layer |
| $H, \Delta H$ | the aggregated entropy value and the entropy variation rate |
| $N, L$ | the number of heads and layers in the model |
| $A$ | the probing intermediate answer $[a_1, a_2, \ldots, a_n]$ |
| $\mathcal{C}$ | the model confidence computed by token probabilities |
| $P, T$ | the input prompt and the generated thoughts |
| $I, J$ | the probing prompt and the jump prompt |
| $Logits_r$ | the normalized logits probability of the intermediate answer in $A$ |
| $A_{global}^W$ | the global attention weight computed following Equation (8) |

## D. Details of Experimental Setup in Section 5

### D.1. Benchmarks

We conduct systematic and comprehensive evaluations on six mathematical reasoning, one scientific reasoning, and two code reasoning benchmarks.

Among the mathematical reasoning benchmarks, GSM8K (Cobbe et al., 2021), MATH-500 (Hendrycks et al., 2021), and AMC 2023 (AI-MO, 2024) are relatively simple, on which many LRMs achieve reasoning accuracies beyond 80–90%. In contrast, AIME 2024 (MAA Committees), AIME 2025 (MAA Committees), and OlympiadBench (He et al., 2024) are considerably more challenging.

- GSM8K consists of 1,319 grade-school–level math word problems with natural language descriptions. Each problem requires the model to extract key information from the textual question and derive a final numerical answer through multi-step arithmetic and logical reasoning. The problems cover basic mathematical concepts such as addition, subtraction, multiplication, division, ratios, time, and monetary values, but typically require multiple intermediate steps to solve, thereby placing moderate demands on the model's CoT reasoning ability.

- MATH-500 is an evaluation subset derived from the MATH dataset, consisting of 500 representative mathematical problems. The problems span multiple mathematical domains, including algebra, geometry, number theory, and combinatorics, and typically require deep reasoning chains and rigorous intermediate steps to arrive at the correct answer.

- AMC 2023 contains 40 mathematical problems covering algebra, geometry, number theory, and combinatorics. Although each problem ultimately requires a single numeric answer, solving them typically involves multiple steps of logical reasoning, symbolic computation, and non-trivial mathematical insights, placing high demands on the model's reasoning coherence and problem decomposition abilities.

- AIME 2024 and AIME 2025 datasets are derived from the American Invitational Mathematics Examination (AIME), organized by the Mathematical Association of America (MAA). AIME serves as one of the advancement examinations in the AMC series and places high demands on mathematical reasoning and problem-solving skills. Each year, AIME consists of two contests (AIME I and AIME II), each comprising 15 fill-in-the-blank problems covering algebra, geometry, number theory, combinatorics, and other domains requiring deep reasoning. AIME 2024 and AIME 2025 each contain 30 problems and their solutions from AIME I and AIME II exams. Problems in both datasets require multi-step mathematical reasoning and logical decomposition, making them widely used to evaluate the performance of large language models on highly challenging mathematical tasks.

- OlympiadBench is a bilingual benchmark designed to evaluate the scientific reasoning abilities of large models at an Olympiad level, challenging state-of-the-art language and multimodal models with highly difficult mathematics and physics problems. The dataset comprises approximately 8,476 problems sourced from Olympiad-level mathematics and physics competitions as well as China's Gaokao exams, with expert step-by-step reasoning annotations provided for each problem to enable detailed analysis of model reasoning and comprehension. In our experimental evaluation, we select the same subset of 675 samples in LIMO (Ye et al., 2025) and DEER (Yang et al., 2025b), allowing for direct rule-based evaluation of the generated answers.

For the scientific reasoning benchmark, GPQA Diamond is the most challenging subset of the GPQA (Graduate-Level Google-Proof Q&A) benchmark, designed to evaluate LLMs on advanced natural science reasoning tasks. While the full GPQA benchmark spans multiple expert-level domains, the Diamond subset specifically collects the hardest questions, typically covering STEM fields such as physics, chemistry, and biology. Each question has been validated by multiple domain experts, and even cross-disciplinary experts rarely achieve perfect scores, making it a rigorous test of reasoning ability. The subset contains 198 multiple-choice questions, each with a standard answer (sometimes accompanied by detailed explanations), allowing evaluation of models' performance in high-level scientific reasoning, knowledge integration, and cross-domain logical deduction. GPQA Diamond has become a key benchmark for assessing state-of-the-art LLMs in complex, academic reasoning scenarios, highlighting both their strengths and limitations.

For the code reasoning benchmarks, HumanEval (Chen et al., 2021) and LiveCodeBench (Jain et al., 2025) are selected.

- HumanEval is a dataset consisting of a series of Python programming problems. Each problem includes a function signature, a descriptive specification, and several unit tests. Models are required to generate a complete code implementation that passes all the tests based on the problem description. The problems range from beginner to intermediate difficulty and are accompanied by executable tests, enabling strict verification of the generated code against the intended functionality. HumanEval evaluates multiple programming abilities, including semantic understanding, logical reasoning, control flow, and API usage. As such, it has become a standard benchmark for assessing large language models on function-level code generation and reasoning capabilities, and it has served as the foundation for subsequent datasets, such as MBPP and HumanEval+.

- LiveCodeBench is a dynamic benchmark designed to comprehensively evaluate the coding abilities of LLMs, addressing potential data leakage and overfitting issues present in traditional static benchmarks. Unlike fixed datasets such as HumanEval, LiveCodeBench continuously collects high-quality programming problems from major online competition platforms, including LeetCode, AtCoder, and Codeforces, with each problem annotated by its publication date to enable rigorous post-training evaluation. The benchmark covers not only traditional code generation tasks but also code self-repair, code execution, and test output prediction, providing a holistic assessment of model performance on real-world software development challenges. Following DEER (Yang et al., 2025b), our evaluation is based on LiveCodeBench-v5, which contains 880 programming problems collected from May 2023 to January 2025.

### D.2. Implementation Details

All evaluations are conducted in a Zero-shot CoT setting with the following prompt: *"Please reason step by step, and put your final answer within \boxed."* The probing prompt $I$ is set to *"\n\n Final Answer\n\n \boxed"* and the jump prompt $J$ is set to *"Wait, my previous reasoning is not correct. I should adopt a more concise and different approach to reexamine this problem.\n\n"*.

In Section 5.3, we evaluate three jump prompt variants to investigate their impact on reasoning accuracy (Acc) and compression ratio (CR): $J_1$ denotes the default jump prompt $J$; $J_2$ represents *"Wait, my initial reasoning may be incorrect; I need to reanalyze the problem.\n\n"*; $J_3$ corresponds to *"Wait, let me reconsider this problem.\n\n"*.

Experiments are conducted using PyTorch 2.6.0 and Python 3.10. The computational hardware consists of the CPU of two 32-core Intel(R) Xeon(R) @ 2.80GHz and GPU of $8\times$ NVIDIA A800, with CUDA 12.1 utilized for hardware acceleration.

## E. More Experimental Results

Table 6 complements Table 1 by presenting the performance of Qwen3-32B and DeepSeek-R1-Distill-Llama-8B on four mathematical and one science reasoning benchmarks. Table 7 illustrates the LRM performance on the other two mathematical reasoning and two code generation benchmarks.

*Table 6.* Additional results on Qwen3-32B and DeepSeek-R1-Distill-Llama-8B.

| Method | GSM8K | | | MATH-500 | | | AIME 2024 | | | OlympiadBench | | | GPQA Diamond | | | AVG | |
|---|---|---|---|---|---|---|---|---|---|---|---|---|---|---|---|---|---|
| | Acc↑ | Len↓ | CR↓ | Acc↑ | Len↓ | CR↓ | Acc↑ | Len↓ | CR↓ | Acc↑ | Len↓ | CR↓ | Acc↑ | Len↓ | CR↓ | Acc↑ | CR↓ |
| | | | | | | *Qwen3-32B* | | | | | | | | | | | |
| Vanilla | 96.3 | 1,734 | 100% | 94.4 | 4,412 | 100% | 73.3 | 11,325 | 100% | 62.7 | 6,497 | 100% | 65.2 | 6,781 | 100% | 78.4 | 100% |
| ASAG (**ours**) | 96.2 | 825 | 47.5% | 95.2 | 3,320 | 75.2% | 76.7 | 8,469 | 74.8% | 68.3 | 5,064 | 77.9% | 66.2 | 4,079 | 60.2% | **80.5** | **67.1%** |
| | | | | | | *DeepSeek-R1-Distill-Llama-8B* | | | | | | | | | | | |
| Vanilla | 89.1 | 1,514 | 100% | 87.6 | 3,928 | 100% | 40.0 | 13,472 | 100% | 47.3 | 8,416 | 100% | 26.3 | 9,655 | 100% | 58.1 | 100% |
| ASAG (**ours**) | 90.5 | 1,137 | 75.1% | 88.6 | 2,458 | 62.6% | 46.7 | 9,236 | 68.6% | 50.2 | 5,219 | 62.0% | 30.3 | 5,632 | 58.3% | **61.3** | **65.3%** |

*Table 7.* Comparison across LRMs of varying scales. The results are reported on four reasoning benchmarks, including two mathematical reasoning and two code reasoning tasks. The best and the second best results are highlighted in **bold** and underline, respectively.

| Method | AMC 2023 | | | AIME 2025 | | | HumanEval | | | LiveCodeBench | | | AVG | |
|---|---|---|---|---|---|---|---|---|---|---|---|---|---|---|
| | Acc↑ | Len↓ | CR↓ | Acc↑ | Len↓ | CR↓ | Acc↑ | Len↓ | CR↓ | Acc↑ | Len↓ | CR↓ | Acc↑ | CR↓ |
| | | | | | | *Qwen3-4B* | | | | | | | | |
| Vanilla | 87.5 | 7,512 | 100% | 46.7 | 12,774 | 100% | 91.5 | 3,821 | 100% | 64.2 | 8,705 | 100% | 72.5 | 100% |
| ASAG (**ours**) | 90.0 | 5,230 | 69.6% | 53.3 | 11,490 | 89.9% | 92.7 | 1,249 | 32.7% | 65.4 | 5,369 | 61.7% | **75.4** | **63.5%** |
| | | | | | | *Qwen3-8B* | | | | | | | | |
| Vanilla | 87.5 | 8,045 | 100% | 53.3 | 13,041 | 100% | 90.2 | 3,812 | 100% | 64.7 | 8,923 | 100% | 73.9 | 100% |
| ASAG (**ours**) | 92.5 | 4,285 | 53.3% | 60.0 | 11,826 | 90.7% | 92.7 | 1,002 | 26.3% | 66.1 | 4,561 | 51.1% | **77.8** | **55.4%** |
| | | | | | | *Qwen3-14B* | | | | | | | | |
| Vanilla | 92.5 | 7,214 | 100% | 60.0 | 12,451 | 100% | 92.7 | 3,198 | 100% | 72.6 | 8,112 | 100% | 79.5 | 100% |
| ASAG (**ours**) | 95.0 | 4,389 | 60.8% | 63.3 | 11,290 | 90.7% | 93.9 | 1,025 | 32.1% | 74.8 | 4,926 | 60.7% | **81.8** | **61.1%** |
| | | | | | | *Qwen3-32B* | | | | | | | | |
| Vanilla | 95.0 | 7,711 | 100% | 66.7 | 12,398 | 100% | 93.3 | 3,216 | 100% | 74.1 | 7,521 | 100% | 82.3 | 100% |
| ASAG (**ours**) | 97.5 | 4,830 | 62.6% | 66.7 | 9,211 | 74.3% | 93.9 | 1,126 | 35.0% | 75.3 | 4,745 | 63.1% | **83.4** | **58.8%** |
| | | | | | | *DeepSeek-R1-Distill-Qwen-7B* | | | | | | | | |
| Vanilla | 77.5 | 6,841 | 100% | 26.7 | 11,861 | 100% | 78.6 | 5,739 | 100% | 38.4 | 10,516 | 100% | 55.3 | 100% |
| ASAG (**ours**) | 82.5 | 4,312 | 63.0% | 36.7 | 7,547 | 63.6% | 78.6 | 1,257 | 21.9% | 40.3 | 2,945 | 28.0% | **59.5** | **44.1%** |
| | | | | | | *DeepSeek-R1-Distill-Llama-8B* | | | | | | | | |
| Vanilla | 72.5 | 6,482 | 100% | 26.7 | 12,142 | 100% | 80.5 | 6,214 | 100% | 41.5 | 11,029 | 100% | 55.3 | 100% |
| ASAG (**ours**) | 80.0 | 4,087 | 63.1% | 33.3 | 7,721 | 63.6% | 82.3 | 1,465 | 23.6% | 43.8 | 3,314 | 30.0% | **59.9** | **45.1%** |

# F. More Ablation Studies

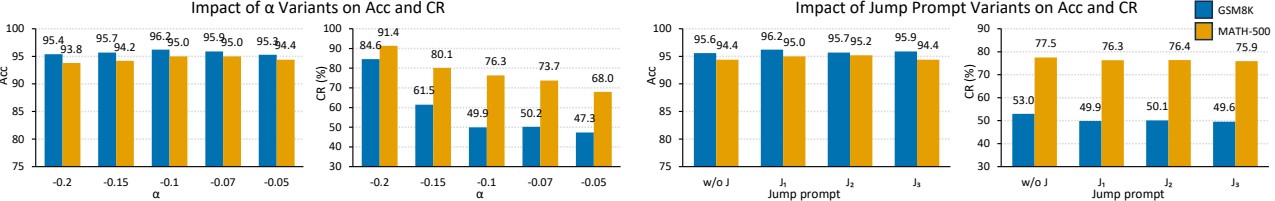

*Figure 7.* Impact of $\alpha$ variants (left) and jump prompt variants (right) evaluated on Qwen3-14B.

More experimental results on $\alpha$ and jump prompt variants are presented in Figure 7, further demonstrating setting $\alpha$ to $-0.1$ achieves a balance between accuracy and efficiency, and that jump prompts can help LRMs escape thinking traps.

# G. Relationship between Attention Entropy and Confidence Signals

To investigate the synergy and potential divergence between internal attention entropy and external confidence signals, we conduct a decoupling analysis between attention entropy $H$ and confidence $\mathcal{C}$.

First, we collect paired $(H, \mathcal{C})$ values at every Action Transition Point (ATP) across AMC 2023 (representing easy tasks) and AIME 2024 (representing challenging tasks) using Qwen3-8B. Confidence $\mathcal{C}$ is calculated as the mean token probability

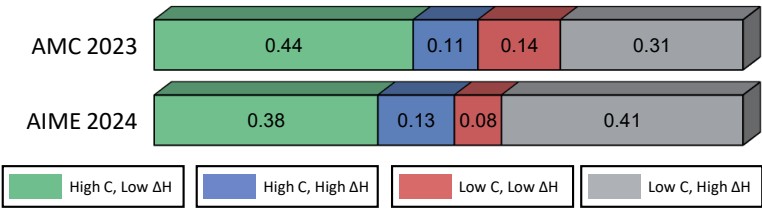

*Figure 8.* Paired $(H, \mathcal{C})$ values at each ATP on AMC 2023 and AIME 2024 evaluated with Qwen3-8B. The numerical values indicate the proportion of samples falling into each category, and the green region denotes converged and reliable reasoning states. High $\mathcal{C}$ and low $\Delta H$ represent $\mathcal{C} > \lambda$ and $\Delta H < \alpha$, respectively.

of the probed intermediate answer following Equation (5), and the entropy variation rate $\Delta H$ is computed as the rate of change in attention entropy between first ATP and current ATP.

We then categorize all ATP observations into four regimes based on the empirical thresholds $\lambda = 0.95$ and $\alpha = -0.1$. The visualization of paired $(H, \mathcal{C})$ values is shown in Figure 8. (1) High $\mathcal{C}$ and low $\Delta H$. High confidence is typically associated with a low entropy variation rate, indicating that as the reasoning process converges, internal attention becomes more concentrated while output confidence increases. In addition, we quantitatively compute the Pearson $\rho$ correlation coefficient between $\mathcal{C}$ and $\Delta H$ on two tasks to roughly examine whether a systematic relationship exists between the two variables. The results show that the Pearson correlation on the relatively easier AMC 2023 ($\rho$=-0.71) is stronger than that on the more challenging AIME 2024 ($\rho$=-0.38). This discrepancy suggests that as task difficulty increases, the alignment between internal stability and external confidence becomes progressively decoupled. (2) High $\mathcal{C}$ and high $\Delta H$. Regarding the ATPs where the LRM exhibit overconfidence, while the token-level logits distribution is sharp, the attention remains diffuse and exploratory indicating that the model's certainty is superficial and lacks a stable evidentiary foundation. In this scenario, ASAG utilizes the high entropy signal as a "stability brake," overriding the deceptive confidence signal to prevent a premature and incorrect early exit. (3) Low $\mathcal{C}$ and low $\Delta H$. This scenario indicates attention convergence, where the LRM has internally formed a stable reasoning trajectory but still exhibits hesitation at token outputs. This behavior may arise because, although attention weights are highly concentrated on key evidence, the LRM retains strong "cognitive inertia" and thus prefers to continue generation rather than committing to a conclusion, resulting in relatively flat output logits. ASAG identifies this state as internal stability and leverages logits injection to guide the model toward a conclusion, thereby substantially improving efficiency without sacrificing accuracy. (4) Low $\mathcal{C}$ and high $\Delta H$. This case corresponds to the slow thinking reasoning stage. At this stage, the model remains in an exploratory state, with attention broadly dispersed to integrate diverse information, and the output remains uncertain. ASAG allows the LRM to continue extended reasoning until a more stable information flow is established. This design ensures that the model is allocated sufficient computational resources to solve complex tasks that require sustained and deliberate reasoning.

## H. Future Work and Discussions

A particularly promising direction for future research is the integration of internal attentional dynamics into the reinforcement learning (RL) framework for training reasoning models. While current large reasoning models (LRMs) primarily rely on outcome-based rewards or sparse rule-based verifiers, our findings suggest that attention dynamics can serve as a robust signal for process-based supervision. Recent work has also shown that manipulating internal representation dynamics can systematically steer model behavior and reasoning trajectories, highlighting the importance of understanding hidden-state evolution during inference (Xu et al., 2026). Future work could explore designing "attention-aware reward functions" that incentivize LRMs to reach reasoning convergence more efficiently. By penalizing redundant reasoning paths where attention remains highly dispersed and rewarding trajectories that exhibit clear entropy drops, LRMs could learn to internalize self-correction and optimal termination strategies during the training phase, fundamentally mitigating "overthinking" at its source rather than relying solely on inference-time interventions. Furthermore, we aim to extend the principles of ASAG to more sophisticated search-based inference algorithms. In frameworks such as Monte Carlo Tree Search (MCTS) or Tree-of-Thought (ToT), attention-state metrics could be utilized as a heuristic for dynamic branch pruning, allowing the system to reallocate computational budget from redundant paths to more promising logical directions. These advancements would move the field closer to "System 2" models that not only possess deep reasoning capabilities but also an internal understanding of their own cognitive progress.

This work also has broad implications across multiple domains. In graph reasoning, GraphCogent (Wang et al., 2026) can leverage our attention-state monitoring to guide reasoning guide through attention distribution analysis. In data distillation, ASAG and AT-BPTT (Li et al., 2026c) share the principle of using internal model dynamics for adaptive computation control. For GraphRAG, CS-RAG (Ma et al., 2026) and NeuroPath (Li et al., 2026a) can integrate our mechanism to eliminate redundant search by determining when sufficient information is retrieved. In adversarial defense (Chen et al., 2026a; Guo et al., 2026), ASAG provides a novel perspective for detecting adversarial examples through anomalous attention patterns and identifying abnormal reasoning states under malicious inputs. These applications demonstrate that attention dynamics represent a general paradigm for assessing model cognitive states beyond isolated optimization.

## I. Theoretical Justification for Incorporating Attention Dynamics

**Lemma I.1** (Entropy–Stability bound). *Consider a single-head self-attention update of a hidden state $h \in \mathbb{R}^d$*

$$h' = \textit{Softmax}(z)\, V, \qquad z = \frac{QK^\top}{\sqrt{d}}, \qquad q = W_Q h,$$

*and denote the attention vector $\alpha = \textit{Softmax}(z)$ with $\alpha_{\max} = \max_i \alpha_i$. Let $J := \frac{\partial z}{\partial h}$. Suppose there exist constants $B_J, B_V > 0$ such that $\|J\|_2 \leq B_J$ and $\|V\|_2 \leq B_V$. Then the Jacobian of the layer satisfies the spectral-norm bound*

$$\|\nabla_h h'\|_2 \leq 2\, \alpha_{\max}(1 - \alpha_{\max})\, B_J B_V.$$

*Moreover, if $H(\alpha)$ denotes the Shannon entropy of $\alpha$ (natural logarithm), $H(\alpha) \to 0$ implies $\alpha_{\max} \to 1$ and hence the right-hand side above tends to 0.*

*Proof.* Write the attention derivative matrix $D := \frac{\partial \alpha}{\partial z} = \text{diag}(\alpha) - \alpha\alpha^\top$. By the chain rule, $\nabla_h h' = D\, J\, V$. Taking operator norms and using submultiplicativity yields

$$\|\nabla_h h'\|_2 \leq \|D\|_2 \|J\|_2 \|V\|_2 \leq \|D\|_2\, B_J B_V.$$

It remains to bound $\|D\|_2$. The entries of $D$ satisfy

$$D_{ii} = \alpha_i(1 - \alpha_i), \qquad D_{ij} = -\alpha_i\alpha_j \ (i \neq j).$$

For any row $i$,

$$\sum_j |D_{ij}| = \alpha_i(1 - \alpha_i) + \sum_{j \neq i} \alpha_i\alpha_j = 2\alpha_i(1 - \alpha_i).$$

Therefore the maximum absolute row sum (the $\infty$-norm) is bounded by $\|D\|_\infty \leq 2\max_i \alpha_i(1 - \alpha_i) = 2\alpha_{\max}(1 - \alpha_{\max})$. By symmetry the same bound holds for the column-sum norm $\|D\|_1$. Using the standard inequality $\|D\|_2 \leq \sqrt{\|D\|_1\|D\|_\infty}$ we obtain

$$\|D\|_2 \leq 2\alpha_{\max}(1 - \alpha_{\max}).$$

Combining the above bounds gives

$$\|\nabla_h h'\|_2 \leq 2\, \alpha_{\max}(1 - \alpha_{\max})\, B_J B_V,$$

which proves the stated spectral-norm bound.

Finally, relate entropy to $\alpha_{\max}$. Since for every $i$ we have $-\log \alpha_i \geq -\log \alpha_{\max}$, the Shannon entropy (natural logarithm) satisfies

$$H(\alpha) = \sum_i \alpha_i(-\log \alpha_i) \geq -\log \alpha_{\max}.$$

Hence $\alpha_{\max} \geq e^{-H(\alpha)}$. Therefore $H(\alpha) \to 0$ implies $\alpha_{\max} \to 1$, and consequently $\alpha_{\max}(1 - \alpha_{\max}) \to 0$, which makes the right-hand side of the spectral bound tend to zero. □

This lemma provides a direct quantitative link between attention concentration and the model's stability. From a reasoning perspective, the reduction of Shannon entropy $H(\alpha)$ serves as a mechanism for state transition: as the distribution concentrates ($\alpha_{\max} \to 1$), the model effectively identifies specific reasoning states by filtering out noise from irrelevant tokens.

Mathematically, this concentration forces the Jacobian factor $D$ to shrink, promoting a global contraction. According to Banach's Fixed-Point Theorem, such a contraction is essential for iterative reasoning, as it ensures the model's internal representations converge toward a stable, unique logical conclusion (a fixed point) rather than oscillating or diverging. Therefore, minimizing entropy is not just a side effect but a structural necessity for the model to lock in on a clear state, thereby enabling robust multi-step inference.

## J. Case Study

**Question**

In a dance class of 20 students, 20% enrolled in contemporary dance, 25% of the remaining enrolled in jazz dance, and the rest enrolled in hip-hop dance. What percentage of the entire students enrolled in hip-hop dance?

## ASAG

Okay, let me try to figure out this dance class problem. So, there are 20 students in total. The question is asking what percentage of the entire students enrolled in hip-hop dance. \n\nFirst, let me break down the information given. \n\n20% of the students enrolled in contemporary dance. Then, 25% of the remaining students enrolled in jazz dance. The rest, which would be the remaining students after contemporary and jazz, enrolled in hip-hop. I need to find the percentage for hip-hop.\n\nAlright, let me start by calculating how many students are in contemporary dance. Since 20% of 20 students are in contemporary, I can calculate that as 20% multiplied by 20. \n\nSo, 20% is the same as 0.20 in decimal form. Multiplying 0.20 by 20 gives me... let me do that. 0.20 * 20 = 4. So, 4 students are in contemporary dance. \n\nNow, the problem says 25% of the remaining students enrolled in jazz dance.

Wait, the remaining students after contemporary. So, total students are 20, subtract the 4 in contemporary, so 20 - 4 = 16 students remaining. \n\nSo, 25% of these 16 students are in jazz. Let me calculate that. 25% is 0.25 in decimal. So, 0.25 * 16. Let me compute that. 0.25 * 16 is the same as 16 divided by 4, which is 4. So, 4 students are in jazz dance. \n\n

Wait, so contemporary is 4, jazz is 4. Then the rest are in hip-hop. So total students in contemporary and jazz are 4 + 4 = 8. Total students are 20, so 20 - 8 = 12 students in hip-hop. \n\nNow, the question asks for the percentage of the entire students who enrolled in hip-hop. So, that would be 12 / 20 * 100. Let me compute that. 12 divided by 20 is 0.6. Multiply by 100 gives 60%. \n\n\n

**</think>**

\n\nTo determine the percentage of students who enrolled in hip-hop dance, we proceed step by step:\n\n---\n\n### **Step 1: Total number of students**\nThe total number of students in the class is:\n$$20$$\n\n---\n\n### **Step 2: Students in contemporary dance**\n20% of the students enrolled in contemporary dance:\n$$20\\% \\text{ of } 20 = 0.20 \\times 20 = 4$$\nSo, **4 students** are in contemporary dance.\n\n---\n\n### **Step 3: Remaining students after contemporary**\nSubtract the number of students in contemporary from the total:\n$$20 - 4 = 16$$\nSo, **16 students** remain.\n\n---\n\n### **Step 4: Students in jazz dance**\n25% of the remaining 16 students enrolled in jazz dance:\n$$25\\% \\text{ of } 16 = 0.25 \\times 16 = 4$$\nSo, **4 students** are in jazz dance.\n\n---\n\n### **Step 5: Remaining students for hip-hop**\nSubtract the number of students in contemporary and jazz from the total:\n$$20 - 4 - 4 = 12$$\nSo, **12 students** are in hip-hop dance.\n\n---\n\n### **Step 6: Percentage of students in hip-hop**\nTo find the percentage of the entire class that enrolled in hip-hop:\n$$\\frac{12}{20} \\times 100 = 60\\%$$\n\n---\n\n### **Final Answer:**\n$$\\boxed{60}$$

## Vanilla

Okay, let me try to figure out this dance class problem. So, there are 20 students in total. The question is asking what percentage of the entire students enrolled in hip-hop dance. \n\nFirst, let me break down the information given. \n\n20% of the students enrolled in contemporary dance. Then, 25% of the remaining students enrolled in jazz dance. The rest, which would be the remaining students after contemporary and jazz, enrolled in hip-hop. I need to find the percentage for hip-hop.\n\nAlright, let me start by calculating how many students are in contemporary dance. Since 20% of 20 students are in contemporary, I can calculate that as 20% multiplied by 20. \n\nSo, 20% is the same as 0.20 in decimal form. Multiplying 0.20 by 20 gives me... let me do that. 0.20 * 20 = 4. So, 4 students are in contemporary dance. \n\nNow, the problem says 25% of the remaining students enrolled in jazz dance.

Wait, the remaining students after contemporary. So, total students are 20, subtract the 4 in contemporary, so 20 - 4 = 16 students remaining. \n\nSo, 25% of these 16 students are in jazz. Let me calculate that. 25% is 0.25 in decimal. So, 0.25 * 16. Let me compute that. 0.25 * 16 is the same as 16 divided by 4, which is 4. So, 4 students are in jazz dance. \n\n

Wait, so now, the remaining students would be those who didn't enroll in contemporary or jazz. So, total students are 20. Subtract the contemporary and jazz students. Let me do that. Contemporary: 4, Jazz: 4. So, total enrolled in contemporary and jazz is 4 + 4 = 8. Therefore, the rest enrolled in hip-hop. So, 20 - 8 = 12 students. So, 12 students enrolled in hip-hop.

**<omitted>**

Wait, so 60% of the students enrolled in hip-hop? Let me check again if that's correct. \n\nAlternatively, maybe I can check the percentages. \n\nContemporary is 20%, jazz is 25% of the remaining.

Wait, but the 25% is of the remaining, not of the total. So, the 25% is applied to the remaining 16 students, not the total 20. So, that's why jazz is 4 students. \n\nSo, then hip-hop is the rest, which is 20 - 4 - 4 = 12. 12/20 is 0.6, which is 60%. \n\nBut let me think again. Maybe there's another way to compute it. \n\nAlternatively, maybe I can compute the percentage for hip-hop directly. \n\nTotal percentage is 100%.

**<omitted>**

Wait, so that seems to confirm the answer. \n\nBut let me check again with numbers. \n\nTotal students: 20. \n\nContemporary: 20% of 20 = 4. \n\nJazz: 25% of (20 - 4) = 25% of 16 = 4. \n\nHip-hop: 20 - 4 - 4 = 12. \n\n12/20 = 0.6 = 60%. \n\nYes, that seems correct.

**<omitted>**

\n\nTherefore, the answer should be 60%.\n\n**Final Answer**\nThe percentage of the entire students who enrolled in hip-hop dance is \\boxed{60}.\n

**</think>**

To determine the percentage of students who enrolled in hip-hop dance, we begin with the total number of students, which is 20.\n\n---\n\n### Step 1: Contemporary Dance Enrollment\n\nWe're told that **20%** of the students enrolled in contemporary dance.\n$$20\\% \\text{ of } 20 = 0.2 \\times 20 = 4 \\text{ students}\n$$\n\nSo, 4 students are in contemporary dance.\n\n---\n\n### Step 2: Remaining Students After Contemporary\n\nSubtract the contemporary dancers from the total:\n$$20 - 4 = 16 \\text{ students remaining}\n$$\n\n---\n\n### Step 3: Jazz Dance Enrollment\n\nNext, **25%** of the **remaining** students enrolled in jazz dance. This is 25% of 16:\n$$25\\% \\text{ of } 16 = 0.25 \\times 16 = 4 \\text{ students}\n$$\n\nSo, 4 students are in jazz dance.\n\n---\n\n### Step 4: Hip-Hop Dance Enrollment\n\nThe rest of the students are in hip-hop dance. Subtract the contemporary and jazz dancers from the total:\n$$20 - 4 - 4 = 12 \\text{ students}\n$$\n\n---\n\n### Step 5: Percentage of Students in Hip-Hop\n\nTo find the percentage of the total students who enrolled in hip-hop:\n$$\\frac{12}{20} \\times 100 = 60\\%\n$$\n\n---\n\n### Final Answer\n\n$$\n\\boxed{60}\n$$

**Token number: 935**

**Token number: 2014**

*Figure 9.* Comparison of the generated results between ASAG and vanilla CoT on a sample from GSM8K. Both methods arrive at the correct final answer; however, ASAG terminates early by evaluating model confidence and attention entropy, thereby significantly improving reasoning efficiency. Blue and yellow parts represent slow thinking stage; green part denotes conclusion stage.

**Question**  How many positive perfect squares less than $2023$ are divisible by $5$?

## ASAG

Okay, so I need to figure out how many positive perfect squares less than 2023 are divisible by 5. Let me start by recalling what a perfect square is. A perfect square is a number that's the square of an integer. So, numbers like 1, 4, 9, 16, etc. Now, the question is asking for those perfect squares that are divisible by 5 and also less than 2023. \n\nFirst, let me think about what it means for a number to be divisible by 5. A number is divisible by 5 if it ends with 0 or 5. But since we're dealing with perfect squares, maybe there's a more mathematical way to approach this rather than just looking at the last digit. \n\nI remember that if a number is divisible by 5, then its prime factorization must include at least one 5. But since we're dealing with perfect squares, all the exponents in their prime factorization must be even numbers. So, if a perfect square is divisible by 5, then 5 must be raised to an even power in its prime factorization. \n\n

Wait, but if a number is divisible by 5, then it must have at least one 5 in its prime factors. However, since it's a perfect square, the exponent of 5 must be even.

**\<omitted\>**
Wait, 45 squared is 2025. So sqrt(2023) is just a bit less than 45. Therefore, the largest integer n such that n² < 2023 is 44, since 44² = 1936, which is less than 2023, and 45² is 2025, which is greater than 2023. \n\n

**\<omitted\>**
So the largest integer multiple is 8*5=40. Therefore, the multiples of 5 are 5, 10, 15, 20, 25, 30, 35, 40. Let me count them. Starting from 5:\n\n5*1=5\n\n5*2=10\n\n5*3=15\n\n5*4=20\n\n5*5=25\n\n5*6=30\n\n5*7=35\n\n5*8=40\n\nSo that's 8 numbers. Therefore, there are 8 multiples of 5 between 1 and 44 inclusive. Therefore, there are 8 perfect squares (each being (5k)^2) less than 2023 that are divisible by 5. \n\n\n

**\</think\>**

To determine how many **positive perfect squares less than 2023** are **divisible by 5**, we proceed with a structured approach:\n\n---\n\n### Step 1: Understand the condition for divisibility by 5\n\nA number is divisible by 5 if it has at least one factor of 5 in its prime factorization. However, since we are dealing with **perfect squares**, the exponents in the prime factorization must all be **even numbers**.\n\nThus, for a perfect square to be divisible by 5, the exponent of 5 in its prime factorization must be **at least 2**. This implies that the **square root** of the number must be divisible by 5. That is:\n\n> If $ n^2 $ is divisible by 5, then $ n $ must be divisible by 5.\n\nSo, we are looking for all **perfect squares** of the form $ (5k)^2 $, where $ k $ is a positive integer, and $ (5k)^2 < 2023 $.\n\n---\n\n### Step 2: Determine the range of $ k $\n\nWe want to find all integers $ k $ such that:\n\n$$\n(5k)^2 < 2023\n$$\n\nSolving for $ k $:\n\n$$\n25k^2 < 2023 \\Rightarrow k^2 < \\frac{2023}{25} = 80.92\n$$\n\n$$\n\\Rightarrow k < \\sqrt{80.92} \\approx 8.996\n$$\n\nSo the largest integer $ k $ satisfying this is $ k = 8 $.\n\n---\n\n### Step 3: Count the valid values of $ k $\n\nSince $ k $ must be a **positive integer**, the possible values of $ k $ are:\n\n$$\nk = 1, 2, 3, 4, 5, 6, 7, 8\n$$\n\nThis gives us **8** values of $ k $, and hence **8** perfect squares of the form $ (5k)^2 $ that are less than 2023 and divisible by 5.\n\n---\n\n### Final Answer\n\n$$\n\\boxed{8}\n$$

**Token number: ~ 1600**

## Vanilla

Okay, so I need to figure out how many positive perfect squares less than 2023 are divisible by 5. Let me start by recalling what a perfect square is. A perfect square is a number that's the square of an integer. So, numbers like 1, 4, 9, 16, etc. Now, the question is asking for those perfect squares that are divisible by 5 and also less than 2023. \n\nFirst, let me think about what it means for a number to be divisible by 5. A number is divisible by 5 if it ends with 0 or 5. But since we're dealing with perfect squares, maybe there's a more mathematical way to approach this rather than just looking at the last digit. \n\nI remember that if a number is divisible by 5, then its prime factorization must include at least one 5. But since we're dealing with perfect squares, all the exponents in their prime factorization must be even numbers. So, if a perfect square is divisible by 5, then 5 must be raised to an even power in its prime factorization. \n\n

Wait, but if a number is divisible by 5, then it must have at least one 5 in its prime factors. However, since it's a perfect square, the exponent of 5 must be even. \n\n

**\<omitted\>**
Wait, 45 squared is 2025. So sqrt(2023) is just a bit less than 45. Therefore, the largest integer n such that n² < 2023 is 44, since 44² = 1936, which is less than 2023, and 45² is 2025, which is greater than 2023. \n\n

**\<omitted\>**
Wait, no, that would be different. Wait, no, if m = n² is divisible by 5, then m = 5k for some integer k. But m is also a perfect square. Therefore, 5k is a perfect square. Therefore, 5 divides the perfect square, so 5 divides the square root. Therefore, n is divisible by 5. So again, n = 5k, so m = 25k². Therefore, 25k² < 2023. So same as before. \n\nTherefore, the answer is 8. \n\nBut maybe I need to check if there's a mistake in the calculation of sqrt(2023).

**\<omitted\>**
Wait, suppose there is a square divisible by 5 but not by 25. Let me suppose that n² is divisible by 5, but not by 25. Then, 5 divides n², but 25 does not divide n². But by the fundamental theorem of arithmetic, if 5 divides n², then 5 divides n.

**\<omitted\>**
Wait, just to make sure, let me check if there are any others. For example, 25, 100, 225, 400, 625, 900, 1225, 1600. \n\nNext would be 2025, which is 45², which is over 2023. So yes, 8 numbers. \n\nTherefore, the answer is 8. \n\nBut I want to make sure that the question didn't have a typo or something. The original problem says \"positive perfect squares less than 2023\". So 2023 is not included. \n\nSo, for example, 44²=1936 is less than 2023. 45²=2025 is more than 2023. Therefore, the squares are up to 44². \n\nTherefore, the answer is 8. \n\nTherefore, I think confidently the answer is 8. \n\n**Final Answer**\nThe number of positive perfect squares less than 2023 that are divisible by 5 is \\boxed{8}.\n

**\</think\>**

To determine how many **positive perfect squares less than 2023** are divisible by **5**, we follow a logical and structured approach.\n\n---\n\n### **Step 1: Understanding the Problem**\nWe are looking for all **perfect squares** $ n^2 $ such that:\n\n- $ n^2 < 2023 $\n- $ n^2 $ is divisible by 5\n\n---\n\n### **Step 2: Key Insight from Number Theory**\nA number is divisible by 5 if and only if it contains the prime factor 5. For a **perfect square** to be divisible by 5, the number itself must be divisible by 5.

**\<omitted\>**
Step 5: Confirm the Squares Are Less Than 2023**\n\nLet's list the squares:\n\n- $ 5^2 = 25 $\n- $ 10^2 = 100 $\n- $ 15^2 = 225 $\n- $ 20^2 = 400 $\n- $ 25^2 = 625 $\n- $ 30^2 = 900 $\n- $ 35^2 = 1225 $\n- $ 40^2 = 1600 $\n\nAll of these are **less than 2023**.\n\nThe next one would be $ 45^2 = 2025 $, which is **greater than 2023**, so it is excluded.\n\n---\n\n### **Conclusion**\n\nThere are exactly **8** positive perfect squares less than 2023 that are divisible by 5.\n\n$$\n\\boxed{8}\n$$

**Token number: ~ 9100**

*Figure 10.* Comparison of the generated results between ASAG and vanilla CoT on a sample from AMC 2023. Compared to the example in Figure 9, this problem is moderately more challenging and requires a greater number of reasoning steps, exhibiting a larger efficiency gap between ASAG and the vanilla model.

**Question**

From an unlimited supply of 1-cent coins, 10-cent coins, and 25-cent coins, Silas wants to find a collection of coins that has a total value of $ N $ cents, where $ N $ is a positive integer. He uses the so-called **greedy algorithm**, successively choosing the coin of greatest value that does not cause the value of his collection to exceed $ N $. For example, to get 42 cents, Silas will choose a 25-cent coin, then a 10-cent coin, then 7 1-cent coins. However, this collection of 9 coins uses more coins than necessary to get a total of 42 cents; indeed, choosing 4 10-cent coins and 2 1-cent coins achieves the same total value with only 6 coins. In general, the greedy algorithm succeeds for a given $ N $ if no other collection of 1-cent, 10-cent, and 25-cent coins gives a total value of $ N $ cents using strictly fewer coins than the collection given by the greedy algorithm. Find the number of values of $ N $ between 1 and 1000 inclusive for which the greedy algorithm succeeds.

## ASAG

Okay, so I need to figure out how many values of N between 1 and 1000 inclusive make the greedy algorithm for making change with 1, 10, and 25 cent coins successful. The greedy algorithm here means that Silas always picks the largest coin possible without exceeding the remaining amount. The question says that the greedy algorithm succeeds for N if there's no other way to make N cents with fewer coins than the greedy method. \n\nFirst, let me make sure I understand the problem. For each N from 1 to 1000, I need to check if the greedy algorithm's coin count is the minimal possible. If it is, then that N counts towards the total. The answer is the number of such Ns.\n\nSo, my task is to find all N in [1, 1000] where the greedy algorithm gives the minimal number of coins. \n\nLet me recall that the greedy algorithm works for certain coin systems. For example, the US coin system (penny, nickel, dime, quarter) is canonical, meaning that the greedy algorithm always gives the minimal number of coins. However, in this problem, we only have 1, 10, and 25 cent coins. Wait, so the US system includes 5-cent coins (nickels), but here we don't have them. So maybe the greedy algorithm isn't always optimal here?
Wait, the problem says that for N=42, the greedy algorithm uses 9 coins (25 + 10 + 7*1), but there's a better way with 6 coins (4*10 + 2*1).
**<omitted>**
**Wait, my previous reasoning is not correct. I should adopt a more concise and different approach to reexamine this problem. \n\n**
**<omitted>**
Wait, no. Wait, the difference was coins(25 + b) - (coins(b) + 1) = -3. Therefore, coins(25 + b) = coins(b) + 1 - 3 = coins(b) - 2. But coins(25 + b) is coins(35) = 3 + 5 = 8. coins(b) + 1 is coins(10) + 1 = 1 + 1 = 2. Therefore, 8 - 2 = 6, not -3. Therefore, my previous derivation must have an error. \n\n
**<omitted>**

**</think>**

**Correct Answer** ✅

\n\nTo determine how many values of $ N $ between 1 and 1000 inclusive make the **greedy algorithm** for making change with 1-cent, 10-cent, and 25-cent coins **optimal**, we need to understand when the greedy approach yields the **minimum number of coins**.\n\n---\n\n### Understanding the Greedy Algorithm\n\nThe greedy algorithm for this coin system works as follows:\n1. Use as many 25-cent coins as possible.\n2. Then use as many 10-cent coins as possible.\n3. Finally, use 1-cent coins for the remainder.\n\nThis approach is **not always optimal**.
**<omitted>**
### Counting the Non-Optimal $ N $\n\nFor each of these 10 values of $ b $, the number of $ N $ between 1 and 1000 such that $ N = 25k + b $ and $ k \\\geq 1 $ is:\n\n$$\n\\\left\\lfloor \\frac{1000 - b}{25} \\\right\\rfloor\n$$\n\nSince $ b \\\in \\{5, 6, ..., 19\\} $, the value $ \\\left\\lfloor \\frac{1000 - b}{25} \\\right\\rfloor = 39 $ for all these $ b $.\n\nThus, each of the 10 values of $ b $ contributes **39** values of $ N $ where the greedy algorithm is **not optimal**.\n\n---\n\n### Final Calculation\n\n- Total number of $ N $ from 1 to 1000: 1000\n- Number of $ N $ where greedy algorithm is **not optimal**: $ 10 \\\times 39 = 390 $\n- Therefore, the number of $ N $ where the greedy algorithm is **optimal** is:\n\n$$\n1000 - 390 = \\\boxed{610}\n$$\n\n---\n\n### ✅ Final Answer\n\n$$\n\\\boxed{610}\n$$

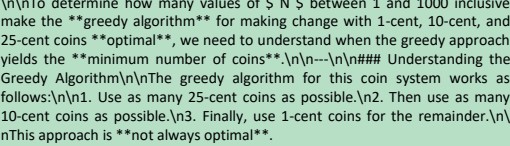

**Token number: ~ 8600**

## Vanilla

Okay, so I need to figure out how many values of N between 1 and 1000 inclusive make the greedy algorithm for making change with 1, 10, and 25 cent coins successful. The greedy algorithm here means that Silas always picks the largest coin possible without exceeding the remaining amount. The question says that the greedy algorithm succeeds for N if there's no other way to make N cents with fewer coins than the greedy method. \n\nFirst, let me make sure I understand the problem. For each N from 1 to 1000, I need to check if the greedy algorithm's coin count is the minimal possible. If it is, then that N counts towards the total. The answer is the number of such Ns.\n\nSo, my task is to find all N in [1, 1000] where the greedy algorithm gives the minimal number of coins. \n\nLet me recall that the greedy algorithm works for certain coin systems. For example, the US coin system (penny, nickel, dime, quarter) is canonical, meaning that the greedy algorithm always gives the minimal number of coins. However, in this problem, we only have 1, 10, and 25 cent coins. Wait, so the US system includes 5-cent coins (nickels), but here we don't have them. So maybe the greedy algorithm isn't always optimal here?
Wait, the problem says that for N=42, the greedy algorithm uses 9 coins (25 + 10 + 7*1), but there's a better way with 6 coins (4*10 + 2*1).
**<omitted>**
Wait, for example, take 25 + b = 25. Then, floor(25 / 10) = 2, remainder 5. So 2 + 5 = 7 coins. But if we use 2*10 + 5*1, that's 7 coins. However, if we could use a 25, but we can't. So, 7 coins is the minimal. However, if we have 25 + b = 30, then floor(30 / 10) = 3, remainder 0. So 3 coins. Which is optimal. Similarly, 25 + b = 35: floor(35 / 10) = 3, remainder 5. So 3 + 5 = 8 coins. But 3*10 + 5*1 = 8 coins. Alternatively, is there a better way? No, since 10s are the next coin. So, yes, that's minimal.\n\nBut in the case of N = 42, which is 25*1 + 17. So, a = 1, b = 17. Greedy coins: 1 + floor(17/10) + 7 = 1 + 1 + 7 = 9. If we take a - 1 = 0 25s, then the amount is 25 + 17 = 42. Then, coins(42) with 10s and 1s is floor(42 / 10) + 42 mod 10 = 4 + 2 = 6 coins. So, total coins would be 0 + 6 = 6, which is better. Therefore, in this case, the greedy algorithm is not optimal.\n\nTherefore, the problem arises when replacing one 25 with 10s and 1s gives fewer coins. Therefore, for each N, we need to check if replacing one 25 (i.e., using a - 1 25s and then making the rest with 10s and 1s) gives fewer coins. If so, then the greedy algorithm is not optimal. Otherwise, it is.\n\n
**<omitted>**
But floor(x) + x mod 10 = floor(x) + (x - 10*floor(x)) = x - 9*floor(x). \n\nWait, no. Wait, for any x, floor(x) + (x mod 10) is not necessarily equal to x - 9*floor(x). \n\nWait, let me take x = (5 + b)/10. \n\nWait, no, let me think differently. \n\nLet me denote x = 5 + b. \n\nThen, floor(x / 1) + (x mod 10)? No, I think I'm getting confused. \n\nWait, coins(c) = floor(c / 10) + (c mod 10). \n\nBut c = 25 + b. \n\nLet me think of c as 25 + b. \n\nLet me write c = 10 * q + r, where 0 ≤ r < 10. \n\nThen, coins(c) = q + r. \n\nBut q = floor(c / 10), r = c mod 10. \n\nBut how does this relate to b? \n\nSince c = 25 + b, and b is between 0 and 24, c is between 25 and 49. \n\nSo, q = floor((25 + b)/10). \n\nFor example, if b=0, c=25, q=2, r=5. \n\nIf b=10, c=35, q=3, r=5. \n\nIf b=15, c=40, q=\n

**</think>**

**Wrong Answer** ❌

\n\nTo determine how many values of $ N $ between 1 and 1000 inclusive make the **greedy algorithm** for making change with 1-cent, 10-cent, and 25-cent coins **optimal**, we need to understand when the greedy approach yields the **minimum number of coins**.\n\n---\n\n### **Understanding the Greedy Algorithm**\n\nThe greedy algorithm for this coin system works as follows:\n\n
**<omitted>**
\n\n### **Counting the Valid Values of $ N $**\n\nWe now count how many values of $ N $ between 1 and 1000 **do not** fall into the non-optimal remainder ranges.\n\n- Total values of $ N $: 1000\n- For each $ q $ from 0 to 39 (since $ 25 \\\times 40 = 1000 $), we consider $ N = 25q + r $, where $ r \\\in [0, 24] $\n- For each $ q $, there are 25 values of $ r $\n- The non-optimal remainders are $ r \\\in [5, 9] \\\cup [15, 19] $, which is 5 + 5 = 10 values per $ q $\n\nSo, for each $ q $, there are 10 values of $ r $ for which the greedy algorithm is **not optimal**.\n\n- Total non-optimal values: $ 40 \\\times 10 = 400 $\n- Total values of $ N $: 1000\n- Therefore, the number of values for which the greedy algorithm **is optimal** is:\n\n$$\n1000 - 400 = \\\boxed{600}\n$$\n\n---\n\n### ✅ Final Answer\n\n$$\n\\\boxed{600}\n$$

**Token number: 16000**

*Figure 11.* Comparison of the generated results between ASAG and vanilla CoT on a sample from AIME 2025. AIME 2025 is one of the most challenging benchmarks. ASAG detects that the LRM becomes stuck in prolonged reasoning, i.e., a thinking trap, and accordingly adds a jump prompt during the reasoning process. This intervention enables the model to reach the correct answer with substantially fewer generated tokens, whereas the vanilla model produces an incorrect result. We hypothesize that, during extended reasoning, the model may have already accumulated partial progress; when further reasoning fails to yield new insights, an early and timely interruption can be beneficial for generating the correct answer.

