# OpenReview forum: "Stop When Further Reasoning Won’t Help: Attention-State Adaptive Generation in Reasoning Models"
_ICML.cc/2026/Conference — ICML 2026 spotlight_

### Official Review · Reviewer_iSCq · 2026-03-02

**Soundness:** 2
**Presentation:** 3
**Significance:** 2
**Originality:** 3
**Overall Recommendation:** 4
**Confidence:** 4

**Summary:**

This paper proposes a training-free inference time framework ASAG for LRM. The core idea is to monitor attention entropy and model confidence at action transition points to determine whether the inference process has converged. Based on these signals, ASAG adaptively applies early stopping, logits guidance, or jump cues to mitigate overthinking and overconfidence. Experiments on multiple inference benchmarks demonstrate that ASAG improves accuracy and significantly reduces generation time on several LRM backbone models.

**Compliance With Llm Reviewing Policy:**

Affirmed.

**Final Justification:**

I think this paper should be accepted by ICML

**Key Questions For Authors:**

See in weakness.

In conclusion, I believe there are problems with the author's motivation and experimental design. If the author can clarify my concerns, especially regarding the design of the prompt, I will raise my score.

**Limitations:**

yes

**Strengths And Weaknesses:**

## Strengths

- The paper addresses a practically important issue in LRMs.

- The paper is very clear, and the figures are easy to understand.

- The paper provides extensive experimental evidence to demonstrate the performance of the method.




## Weakness

- The authors emphasize that outputting words like 'Wait' in the model is considered a switching of the reasoning action, and use this as a key anchor point for this work. However, for many methods, such as simple problems like GSM8K, the model does not actively output words like 'Wait'. If the authors designed additional prompts, they should explain this in the paper; this is important.

- As I mentioned before, I believe that forcing the model to generate 'Wait' would significantly increase the length of the model's response. I see in Table 1 that the length of the vanilla model in GSM8K is 2142. For such simple data as GSM8K, I don't think the model would generate such a long CoT if it sampled regular prompts. Therefore, I have doubts about the authors' experimental setup. If a prompt is used to force the model to output 'wait', and then an early exit occurs in the middle, I don't think this demonstrates the superiority of ASAG.

- While the authors obtained positive results after using the probing prompt, I believe a more reasonable approach to calculating the entropy would be after outputting words like "wait." If the authors calculate the entropy after the probing prompt, it's impossible to determine whether the decrease in entropy is due to the model already having a definite answer or the result of the prompt.

- I acknowledge the richness of the authors' experiments, but using only Qwen or Deepseek (distilled from Qwen) as the benchmark significantly reduces the method's generalization ability. Adding Llama3.1 or gemma3-it as comparative experiments is necessary.

- Regarding hyperparameter selection, the authors only conducted ablation experiments on α, but did not provide detailed explanations for the choices of the confidence threshold and the values ​​of 0.95 and 0.05 in the logits inject.

---

> ### Author Rebuttal · Authors · 2026-03-31
>
> We appreciate the reviewer's observation regarding the prompt design, the confusion about long CoT, the method design, the experiments on other LLMs, and hyperparameter configurations.
>
> ### *W1: Prompt Design*
>
> **We detail the prompt configuration in Appendix D.2.** Specifically, we adopt a zero-shot CoT setting for all experiments: the system prompt is "Please reason step by step, and put your final answer within \\boxed.", and the user prompt is the original question. This setup is consistent with our code.
>
> Importantly, the prompt does **not enforce the generation of markers such as "wait"**. Instead, such markers **naturally emerge in the reasoning process of LRMs,** even on simple tasks like GSM8K. We analyze ATP occurrences across different LRMs, and the results indicate that this behavior is intrinsic to LRMs rather than induced by our prompt.
>
> Please refer to the response to Reviewer mbaZ's ***W2*** for details.
>
> ### *W2: Confusion about Long CoT*
>
> As noted in ***W1***, the prompt **does not enforce the generation of ATPs.** The long outputs (e.g., token length 2142 on GSM8K) are primarily due to the overthinking phenomenon. Dynasor [1] shows that even after reaching a confident answer, LRMs often continue additional steps such as double checking, completeness checking, reassessment, and confidence building.
>
> Therefore, the long reasoning traces reflect the model's intrinsic generation behavior, **rather than arising from manual experimental setup.** The reduction achieved by ASAG demonstrates its ability to detect convergence and mitigate unnecessary reasoning, which directly targets this overthinking phenomenon.
>
> [1] Fu Y, et al. Efficiently scaling llm reasoning with certaindex. NeurIPS, 2025.
>
> ### *W3: Method Design*
>
> We find that computing entropy directly on the original generation (e.g., immediately after tokens like "wait") **does not yield clear or consistent entropy drops, even when reasoning has converged.** We attribute this to the variability of subsequent reasoning actions after such tokens (discussed in ***W2***), which results in unstable entropy signals.
>
> To obtain a more reliable estimate, we compute entropy within a decoding window covering the probing prompt $I$ and an equal number of recent tokens from the generated reasoning $T$ (Section 4, Equation (6)). This ensures that entropy is always measured **under a consistent context across all ATPs, making the resulting entropy variation comparable and preventing it from being affected by the prompt.** Importantly, the probing prompt is used as a tool to detect the model's reasoning state and confidence, without influencing the calculation of entropy variation.
>
> ### *W4: Experiments on Other LLMs*
>
> Our work focuses on mitigating the **overthinking phenomenon in LRMs**, which typically exhibit long and redundant reasoning traces (Section 3.1). In contrast, standard LLMs (e.g., Llama3.1 or gemma3-it) tend to directly produce answers without extensive intermediate reasoning.
>
> As shown in Table 1, we conduct preliminary experiments on gemma3-4b-it with the same zero-shot CoT prompt setting (***W1***). Compared to an LRM of similar scale (Qwen3-4B), this model achieves both **substantially lower accuracy and shorter token lengths**. This gap indicates that the overthinking phenomenon is much less pronounced in such models, placing them **outside the primary scope of our method.**
>
> For model generalization, **we additionally evaluate ASAG on the DeepSeek-LLaMA series (Appendix E),** which involves a different backbone architecture, and observe consistent improvements. This indicates that our method generalizes across different LRM variants.
>
> Table 1 Results on gemma3-4b-it and Qwen3-4B.
>
> |Model|GSM8K|GSM8K|MATH-500|MATH-500|AIME 2025|AIME 2025|GPQA-D|GPQA-D|AVG|AVG|
> |-|-|-|-|-|-|-|-|-|-|-|
> ||Acc|Len|Acc|Len|Acc|Len|Acc|Len|Acc|Len|
> |gemma3-4b-it|88.0|397|57.2|1188|6.7|4333|29.3|996|45.3|1729|
> |Qwen3-4B|93.8|2142|92.4|4910|46.7|12774|46.5|9277|69.9|7276|
>
> ### *W5: Hyperparameter Configurations*
>
> (1) $\lambda$: The confidence threshold $\lambda$ is grounded in the **confidence polarization phenomenon**, where the model tends to exhibit extremely high probability to answers once it becomes sufficiently confident. Please refer to the response to Reviewer mbaZ's ***W1*** for details.
>
> (2) 0.05 for $logits_r$: The logits injection coefficient controls the strength of guidance from intermediate answers. In practice, we observe that performance is **stable within a small range of coefficients**, indicating low sensitivity to this hyperparameter. A small value (e.g., 0.05) provides effective guidance without overriding the model's original generation, while larger values overly bias the model toward intermediate predictions and degrade performance.  Therefore, 0.05 is selected as a representative value within this stable region, achieving a good balance between guidance strength and efficiency, without requiring hyperparameter sweeping.

---

> > ### Author Rebuttal · Reviewer_iSCq · 2026-04-01
> >
> > authors solve my concerns, I have raised my score.

---

> > > ### Author Response · Authors · 2026-04-02
> > >
> > > Thank you for your careful review and constructive feedback. Your comments have helped us substantially improve the paper, particularly by elaborating on the prompt design, explaining the intrinsic nature of long CoTs, describing our method design, outlining our method scope, and clarifying the hyperparameter configurations. These revisions strengthen both the empirical support and the clarity of our method. We sincerely appreciate your insightful suggestions, which have made the revised paper more rigorous and easier to follow.

---

### Official Review · Reviewer_gHML · 2026-03-09

**Soundness:** 4
**Presentation:** 4
**Significance:** 3
**Originality:** 3
**Overall Recommendation:** 5
**Confidence:** 4

**Summary:**

The paper addresses the problem of "overthinking" in reasoning language models (LRMs), where the reasoning output is often longer than necessary, leading to increased computational load, inference inefficiencies, and reduced accuracy. To alleviate these issues, the authors suggest breaking the reasoning process into multiple actions. The transitions between these actions are indicated by action transition points (ATPs), which are marked by specific tokens such as "wait" or "hmm." The core hypothesis is that as the reasoning process converges toward the correct answer, the model's attention entropy decreases. Building on this, the method uses the model's confidence and the entropy of the attention weights at the ATPs to guide generation, either by terminating the process early or by adjusting the logits to steer the model. Unlike prior approaches, this training-free method avoids over-reliance on raw LLM confidence, which is often miscalibrated (too high for hard problems and too low for easy ones). Through extensive simulations, the paper first verifies the attention entropy reduction hypothesis and then demonstrates the overall effectiveness of the method.

**Compliance With Llm Reviewing Policy:**

Affirmed.

**Final Justification:**

The authors rebuttal has addressed my concerns and as such I have decided to raise my score. I find the paper to be impactful, well written and technically sound.

**Key Questions For Authors:**

Please see the detailed points raised in the Weaknesses section. I will consider raising the overall score if the appropriate experiments are added during the rebuttal or if a clear, convincing explanation is provided as to why they are not necessary.

**Limitations:**

yes

**Strengths And Weaknesses:**

## Strengths:

* Important Problem: Addresses the critical issue of "overthinking" in LRMs, which causes computational inefficiency and degrades accuracy.

* Original Method: Proposes an innovative, training-free approach using attention entropy at Action Transition Points (ATPs) to avoid reliance on miscalibrated LLM confidence scores.

* Well-Designed Experiments and Strong Results: Effectively verifies the core hypothesis through simulation and empirically demonstrates significant reductions in computational overhead while maintaining reasoning accuracy.

* Clear Presentation: Well-structured and clearly written, offering a logical flow from the theoretical hypothesis to practical implementation.

## Weaknesses:

**W1 - Statistical Rigor**: The evaluation lacks confidence intervals, as the experiments were conducted using only a single repetition. This severely limits the ability to ascertain the statistical significance, variance, and overall reliability of the reported results.

**W2 - Incomplete Baselines for Deepseek-R1-Distill-Llama-8B**: The evaluation on this model only includes a comparison to the vanilla baseline. Omitting the other established baselines obscures the method's true relative advantage on this architecture.

**W3 - Unjustified Parameters and Missing Ablations**: The paper lacks critical ablation studies to isolate the effectiveness of its specific components. First, it is unclear whether incorporating "jump" attempts actually affects performance; an ablation varying the number of permitted jumps—including a zero-jump baseline—is necessary. Second, the adjusted logits are a mixture of two components, the paper does not adequately explain how logits mixing parameters are determined, leaving the method's sensitivity to these hyperparameters untested.

**W4 - Generalization of the Core Hypothesis**: While the attention entropy reduction hypothesis is successfully verified in simulation, the paper lacks validation across diverse model architectures or scales. It is currently unclear if this entropy behavior is a universal trait of LRMs or an artifact of the specific models tested.

---

> ### Author Rebuttal · Authors · 2026-03-31
>
> We thank the reviewer for the thoughtful comments regarding statistical rigor, incomplete baselines, unjustified parameters and missing ablations, generalization of the core hypothesis.
>
> ### *W1: Statistical Rigor*
>
> In our experiments, we adopt **greedy decoding** (Section 5.1), which leads to deterministic generation. Given the same input, the model produces identical outputs across multiple runs, resulting in zero variance. This deterministic setting ensures strict reproducibility and fair comparison across all baselines.
>
> We conduct further experiments across several datasets and LRMs under a sampling setting (top_p=0.95, temperature=0.6). For each instance, **we perform three sampling runs and report the averaged results to ensure stability and reliability.** Table 1 shows that ASAG is not restricted to deterministic decoding; it consistently maintains its effectiveness under sampling, achieving robust improvements in both accuracy and token efficiency across various LRMs and datasets.
>
> Table 1 Performance under the sampling setting.
>
> |Model|Method|GSM8K|GSM8K|MATH-500|MATH-500|AIME 2025|AIME 2025|AVG|AVG|
> |-|-|-|-|-|-|-|-|-|-|
> |||Acc|Len|Acc|Len|Acc|Len|Acc|Len|
> |Qwen3-4B|Vanilla|94.1|2307|92.6|5082|47.8|13670|78.2|7020|
> ||ASAG|94.0|1158|92.9|3315|53.3|11521|**80.1**|**5331**|
> |Qwen3-8B|Vanilla|95.6|2276|92.7|5250|54.4|14367|80.9|7298|
> ||ASAG|95.8|944|93.1|3184|58.9|12098|**82.6**|**5409**|
> |Deepseek-Llama-8B|Vanilla|89.6|1651|88.5|4439|30.0|12753|69.4|6281|
> ||ASAG|90.4|1162|88.7|2523|32.2|7884|**70.4**|**3856**|
>
> ### *W2: Incomplete Baselines*
>
> We extend our experiments on Deepseek-Llama-8B to include all established baselines. As shown in Table 2, ASAG consistently demonstrates a clear advantage over both prompt-based and output-based methods. Notably, compared to the strongest baseline DEER, ASAG **achieves a higher accuracy with significantly fewer tokens (4237 vs. 4900)**.
>
> Table 2 Comparison results on Deepseek-Llama-8B.
>
> |Method|GSM8K|GSM8K|MATH-500|MATH-500|AIME 2025|AIME 2025|GPQA-D|GPQA-D|AVG|AVG|
> |-|-|-|-|-|-|-|-|-|-|-|
> ||Acc|Len|Acc|Len|Acc|Len|Acc|Len|Acc|Len|
> |Vanilla|89.1|1514|87.6|3928|26.7|12142|26.3|9655|57.4|6810|
> |NoThinking|85.3|486|84.4|2029|16.7|7136|23.2|1773|52.4|**2856**|
> |TALE|88.4|1079|87.4|3385|26.7|11754|26.3|5926|57.2|5536|
> |Dynasor|90.1|1310|88.6|3427|30.0|10433|25.8|6179|58.6|5337|
> |DEER|90.5|1266|88.2|2763|33.3|9252|28.8|6320|60.2|4900|
> |ASAG|90.5|1137|88.6|2458|33.3|7721|30.3|5632|**60.7**|4237|
>
> ### *W3: Unjustified Parameters and Missing Ablations*
>
> (1) $s$: As discussed in Section 5.3, introducing jump prompts **improves accuracy while reducing token length** compared to the zero-jump baseline. However, increasing $s$ doesn't bring further gains; it typically results in **longer generations with little to no improvement in accuracy**. This is because the model tends to follow similar reasoning paths even after multiple jump attempts, causing redundant reasoning rather than effective exploration.
>
> (2) 0.05 for $logits_r$: The logits injection coefficient controls the strength of guidance from intermediate answers. In practice, we observe that performance is **stable within a small range of coefficients**, indicating low sensitivity to this hyperparameter. A small value (e.g., 0.05) provides effective guidance without overriding the model's original generation, while larger values overly bias the model toward intermediate predictions and degrade performance. Notably, when the coefficient is set to 0, task accuracy remains comparable while token length increases (Section 5.3), indicating that this mechanism improves efficiency **without relying on precise tuning**.
>
> ### *W4: Generalization of the Core Hypothesis*
>
> (1) Empirical Validation: We extend the analysis in Section 3 to additional LRMs, **including Qwen3-14B and DeepSeek-Llama-8B**. We observe consistent entropy dynamics across these models: when reasoning converges, entropy exhibits a clear reduction, whereas before convergence, entropy remains unstable. **This indicates that the observed behavior is not specific to a particular model.** Our analysis of the entropy variation distribution also provides strong support for this hypothesis; please refer to the response to Reviewer mbaZ's ***W1*** for details.
>
> (2) Theoretical Universality: This consistency is not coincidental. As theoretically justified in Appendix I, the reduction in Shannon entropy is a fundamental mechanism of state transition in the Transformer's self-attention layers. For reasoning to stabilize on a reliable conclusion, the attention distribution must **concentrate to filter out irrelevant token noise**, mathematically shrinking the Jacobian factor and promoting global contraction.
>
> Therefore, both the empirical observations and theoretical formulations confirm that this entropy behavior is **a universal trait of LRMs, rather than an architecture-specific artifact**. We will incorporate these analyses into the revised manuscript.

---

> > ### Author Rebuttal · Reviewer_gHML · 2026-04-01
> >
> > Thank you for addressing my concerns, I have raised my score.

---

> > > ### Author Response · Authors · 2026-04-02
> > >
> > > Thank you for your careful review and constructive feedback. Your comments have helped us substantially improve the paper, particularly by enhancing statistical rigor, broadening the baseline comparisons, clarifying the hyperparameter configurations, and validating the universality of our core hypothesis. These revisions strengthen both the empirical support and the clarity of our method. We sincerely appreciate your insightful suggestions, which have made the revised paper more rigorous and easier to follow.

---

### Official Review · Reviewer_LTV2 · 2026-03-13

**Soundness:** 3
**Presentation:** 2
**Significance:** 3
**Originality:** 2
**Overall Recommendation:** 4
**Confidence:** 3

**Summary:**

This paper introduces “attention-state adaptive generation” (ASAG), a method for controlling early exiting in LRMs to reduce token cost. While previous work has used token-level confidence to trigger early exit, they identify an additional signal from the attention pattern at action transition point (ATP) tokens (e.g., “wait”). They find that, around the position where the model is able to produce the correct answer (determined via probing prompts), the entropy of the attention distribution drops sharply. They therefore use the deltas in the attention entropy as a signal, in addition to token confidence, to trigger early exit. Early exit is forced if the confidence is above a threshold AND the entropy delta is below a threshold. If the entropy delta is below the threshold while confidence is low, they use a logit injection method to increase the probability of the intermediate answer to encourage the model to stop reasoning. They also include a “jump prompt” method that detects when the model repetitively attends to older reasoning steps and forces a change in approach by appending a special prompt.

**Compliance With Llm Reviewing Policy:**

Affirmed.

**Final Justification:**

See rebuttal acknowledgement.

**Key Questions For Authors:**

See weaknesses

**Limitations:**

yes

**Strengths And Weaknesses:**

Strengths:
-This paper identifies a new phenomenon, where the entropy of the attention distribution is correlated with the model’s ability to produce the correct answer at an intermediate step.
- Their method outperforms confidence-only methods for early exit, achieving better accuracy/token cost tradeoff.

Weakness:
-Presentation: section 4 needs to be split into sections. Currently it introduces three different methods with no separation.
- Analysis of the attention entropy phenomenon could be more thorough: for example, what happens in traces with incorrect answers? Do the same conclusions hold for non-ATP tokens?
- The logit injection method is a bit unclear to me. How is the intermediate answer a_r selected? Isn’t this equivalent to outputting a_r with some probability and allowing the model to continue and produce an (unperturbed) answer otherwise? Could you alternatively increase the logits corresponding to the probing prompt I so that the model can still choose the answer upon stopping?
- How would the hyperparameter alpha be selected in practice? Can the optimal boundary be estimated from distributions like those in figure 2, or do we have to sweep over a range of them to find the best?

---

> ### Author Rebuttal · Authors · 2026-03-31
>
> We thank the reviewer for the valuable insights regarding the method presentation, further analysis on attention entropy, the details of logits injection operation, and hyperparameter configurations.
>
> ### *W1: Method Presentation*
>
> Thank you for the suggestion. We agree that clearer structural separation would improve readability, and we will revise Section 4 by organizing the method into distinct subsections.
>
> ### *W2: Further Analysis on Attention Entropy*
>
> (1) Traces with incorrect answers: We conduct further experiments to analyze entropy variation in incorrect samples. We observe that **entropy variation in these cases tends to be irregular, without clear decreasing trends.** This phenomenon aligns with our theoretical interpretation: the model struggles to form a stable reasoning path and fails to reach a converged reasoning state in incorrect samples, resulting in persistent fluctuating attention entropy.
>
> (2) Non-ATP tokens: While entropy variation can be observed throughout the generated sequence, the signal at non-ATP tokens is dominated by **local token-level fluctuations** (e.g., focusing on syntax or finish constructing a word \[1]), making it much noisier and less indicative of reasoning state transitions. In contrast, ATPs mark the natural boundaries of thought actions in the reasoning process, where entropy changes are **more pronounced and reliable** for determining reasoning convergence. We will include the additional analyses in the revised manuscript.
>
> [1] Wang S, et al. Beyond the 80/20 rule: High-entropy minority tokens drive effective reinforcement learning for llm reasoning. NeurIPS, 2025.
>
> ### *W3: Details of Logits Injection Operation*
>
> When encountering an ATP, ASAG enters the probing stage, and $a_r$ denotes the intermediate answer token generated in this stage. **The logit injection mechanism is not equivalent to probabilistically choosing between outputting $a_r$ and continuing generation.** Instead, it serves as a **soft posterior regularization**, gently biasing the model's subsequent token distribution toward the converged answer $a_r$. This provides a **continuous guidance signal rather than a discrete decision**, enabling the model to break its "cognitive inertia" and commit to a stable conclusion earlier without abruptly truncating its reasoning trajectory.
>
> Regarding the alternative of increasing the logits of the probing prompt $I$, such an approach would encourage the model to transfer to the conclusion stage at arbitrary, non-ATP positions. However, **ATPs represent the natural boundaries of "thought actions" where the model has formed a partial conclusion.** Increasing probing frequency outside of ATPs would force the model to probe for an answer before its internal reasoning state has stabilized. Consequently, the signals at these non-ATP locations are highly **unreliable and noisy**, resulting in premature and incorrect terminations. ASAG avoids this by intervening only at meaningful ATP boundaries, providing more targeted and fine-grained control.
>
> ### *W4: Hyperparameter Configurations*
>
> The hyperparameter $\alpha$ can be reliably selected in a **data-driven and robust manner from empirical distributions**, without the need for extensive grid search or hyperparameter sweeping.
>
> As you noted, the optimal boundary can be inferred from the entropy dynamics (similar to Figure 2). Through additional analysis, we identify a clear **polarization phenomenon in entropy variation** ($\Delta H$) across different models and tasks. Specifically, when we examine $\Delta H$ at all ATPs, the values mainly fall into two extreme intervals: highly converged ($\Delta H \in (-\infty, -0.15)$) and insufficient reasoning ($\Delta H \in [-0.07, \infty)$).
>
> This distribution naturally yields a stable interval for $\alpha$. In practice, we choose $\alpha=−0.1$, which lies in the intermediate region and effectively distinguishes converged from non-converged states. Because this polarization is a shared property of the Transformer's attention convergence, $\alpha=−0.1$ **acts as a universal, robust boundary that transfers across different models and tasks without requiring re-calibration.**
>
> Please refer to the response to Reviewer mbaZ's ***W1*** for a detailed quantitative statistics.

---

> > ### Author Rebuttal · Reviewer_LTV2 · 2026-04-02
> >
> > Thank you for your response. This resolves my questions about the method.
> > However, given the complexity of the method, the improvement from the confidence-only baseline DEER seems marginal (Table 1). The improvement in accuracy and compression rate is less than 2% across the board. Thus, I choose not to raise my score and maintain a weak accept.

---

> > > ### Author Response · Authors · 2026-04-05
> > >
> > > Thank you for your careful review and for confirming that your questions regarding method presentation, entropy analysis, logits injection, and hyperparameter configurations have been resolved. We are deeply grateful for your continued support and positive evaluation of our work.
> > >
> > > Regarding the comparison with DEER, we would like to clarify two points:
> > >
> > > (1) **Implementation Simplicity:** ASAG is well aligned with its underlying motivation and remains lightweight in practice. While it incorporates attention dynamics beyond DEER, it only requires computing entropy from the existing attention cache. Therefore, ASAG introduces no extra parameters and negligible additional computational overhead.
> > >
> > > (2) **Consistent improvements over a strong baseline:** While the absolute gains over DEER may seem incremental, they are **consistent across models, datasets, and evaluation metrics**, including both accuracy and compression rate. Moreover, ASAG introduces a **complementary signal based on attention dynamics**, providing additional, non-redundant information beyond confidence-based approaches for adaptive generation.
> > >
> > > We sincerely appreciate your insightful suggestions, which have made the revised paper more rigorous and easier to follow.

---

### Official Review · Reviewer_mbaZ · 2026-03-21

**Soundness:** 3
**Presentation:** 4
**Significance:** 3
**Originality:** 3
**Overall Recommendation:** 5
**Confidence:** 3

**Summary:**

This paper aims to solve the problems that large reasoning models (LRMs) usually overthinking during inference, which increases token costs and generates redundant reasoning progress. To tackle this problem, this paper proposes Attention-State Adaptive Generation (ASAG), a training-free, output-based framework that adaptively adjusts the reasoning process for efficiency and accuracy. The main idea is to combine the attention entropy with model confidence, and dynamically making decisions on action transition points during reasoning. The experiments shows that ASAG can reduce a lot of token outputs while maintaining the performance across multiple models and benchmarks.

**Compliance With Llm Reviewing Policy:**

Affirmed.

**Final Justification:**

The authors' rebuttal have fully solved my concerns. I'll raise my score to 5 as I think this paper is good.

**Key Questions For Authors:**

Overall, the paper is well-structured. I just have some small problems (can also refer to weaknesses above):
1. How will the ASAG select ATPs if the LRMs don't output "wait" or "hmm" tokens during reasoning?
2. Is there any theoretical support about the selection of hyperparameters? How will the Acc and CR change if increasing the jump limit $s$?
3. How will the ASAG perform under the sampling situation? Does it still work for sampling during reasoning?

**Limitations:**

Yes.

**Strengths And Weaknesses:**

Strengths:
1. The writing of this paper is clear and easy to understand, the motivation behind this work is strong and convincing.
2. The proposed method ASAG is simple but useful. ASAG is a training-free, plug-and-play framework that only adjusts the reasoning process without changing any model weights.
3. The experiments are solid and thorough across multiple models and benchmarks. ASAG gains an advantage over many baselines on math, coding and reasoning areas, which shows its scalability and generalizability.

Weaknesses:
1. The selection of hyperparameters have a solid experiments support, but doesn't have too much theoretical support, like $\lambda = 0.95$, $\alpha = -0.1$, and $0.05$ for $logits_r$. The paper presents the ablation experiments and some analysis on Qwen3-8B and Qwen3-14B with the selection of hyperparameters, and finds that $\alpha = -0.1$ is a "sweet spot", but why do these values have better performance? Is there any theoretical analysis? and does the same hyperparameters perfectly fit in Deepseek-llama models, as they may have a bit different model structures?
2. the paper mentions that ATPs are usually marked by tokens such as "wait", "hmm" or other words like that, which seems to be too specific for all LRMs. Did the previous works consider this questions or how will the method select the ATPs when it comes to other models?

---

> ### Author Rebuttal · Authors · 2026-03-31
>
> We thank the reviewer for the insightful questions regarding the hyperparameter configurations, the selection of ATPs, and our performance under sampling strategy.
>
> ### *W1: Hyperparameter Configurations*
>
> (1) $\lambda$: The confidence threshold $\lambda$ is grounded in the **confidence polarization phenomenon** in LLMs \[1]\[2][3]. Once the model becomes sufficiently confident, it tends to generate the answer with exceptionally high probability (typically exceeding 0.99). Therefore, when $\lambda$ lies within a relatively wide range (e.g., 0.9–0.97), model performance remains robust. We adopt $\lambda=0.95$ without introducing additional sensitivity.
>
> (2) $\alpha$: Appendix I shows that attention shifts from a distributed (high entropy) to a concentrated pattern (low entropy) upon convergence. Due to the softmax-based attention mechanism, this transition is typically **non-linear, leading to sharp entropy drops rather than gradual changes.**
>
> We further analyze the robustness of $\alpha$ and observe a **polarization phenomenon in entropy variation.** We examine entropy variation ($\Delta H$) at all ATPs across models and tasks, and group values into three intervals: highly converged ($\Delta H \in (-\infty, -0.15)$), intermediate ($\Delta H \in [-0.15, -0.07)$), and insufficient reasoning ($\Delta H \in [-0.07, \infty)$). Table 1 shows that most cases lie in the extremes, with few in between. **This makes $\alpha$ insensitive within $[-0.15, -0.07)$, supporting $\alpha=-0.1$  as a robust "sweet spot".** More challenging tasks (e.g., AIME 2025) exhibit more insufficient reasoning, reflecting the need for more reasoning steps.
>
> Table 1 Entropy variation distribution (%) across models and tasks.
>
> |Model|Dataset|$(-\infty, -0.15)$|$[-0.15, -0.07)$|$[-0.07, \infty)$|
> |-|-|-|-|-|
> |Qwen3-8B|GSM8K|64.35|5.66|29.99|
> ||MATH-500|52.83|5.90|41.27|
> ||AIME 2025|17.29|2.78|79.93|
> |Qwen3-14B|GSM8K|67.64|4.71|27.65|
> ||MATH-500|55.22|4.88|39.90|
> ||AIME 2025|20.73|2.04|77.23|
> |Deepseek-Llama-8B|GSM8K|65.02|5.39|29.59|
> ||MATH-500|51.58|5.60|42.82|
> ||AIME 2025|15.76|2.35|81.89|
>
> (3) 0.05 for $logits_r$: This mechanism can be viewed as **posterior regularization**, where the intermediate answer acts as a weak prior guiding the model toward a stable solution more quickly. A small coefficient (0.05) limits this influence, preventing bias toward incorrect answers while **balancing guidance strength and generation quality.**
>
> (4) Generalizability on Deepseek-Llama: As shown in Table 1, Deepseek-Llama exhibits the same entropy polarization as Qwen. Because ASAG relies on these **model-agnostic signals** (attention entropy variation and confidence), our hyperparameters perfectly transfer to Deepseek-Llama despite structural differences, which is empirically validated in Appendix E.
>
> [1] Yang C, et al. Dynamic early exit in reasoning models. ICLR, 2026.
>
> [2] Kadavath S, et al. Language models (mostly) know what they know. arXiv, 2022.
>
> [3] Zhang M, et al. Calibrating the Confidence of Large Language Models by Eliciting Fidelity. EMNLP, 2024.
>
> ### *W2: Selection of ATPs*
>
> The presence of markers like "wait" is a **natural phenomenon in LRMs**  \[1]\[4]\[5]. Table 2 further confirms this by reporting their occurrence on GSM8K under **prompting with only the questions, without extra instructions.** These markers appear consistently across LRMs.
>
> Table 2 Average number of ATP occurrences on GSM8K.
>
> ||wait|alternatively|hmm|
> |-|-|-|-|
> |Qwen3-8B|2.92|4.67|1.45|
> |Qwen3-14B|2.64|3.83|1.58|
> |Deepseek-Llama-8B|2.20|3.29|1.52|
>
> In addition, we replace "wait" with other markers in ASAG and observe similar performance improvements in Table 3, suggesting that ASAG is robust to the choice of markers. **Crucially, structural boundaries like "\n\n" can also serve as ATPs without modifying our framework \[4].**
>
> Table 3 Performance with different ATPs on Qwen3-8B.
>
> ||GSM8K|GSM8K|AIME 2025|AIME 2025|
> |-|-|-|-|-|
> ||Acc|Len|Acc|Len|
> |Vanilla|94.6|2338|53.3|13041|
> |ASAG w/ wait|95.9|1063|60.0|11826|
> |ASAG w/ alternatively|95.7|1147|60.0|11560|
> |ASAG w/ hmm|96.0|1229|60.0|12035|
>
> [4] Yang W, et al. Speculative thinking: Enhancing small-model reasoning with large model guidance at inference time. COLM, 2025.
>
> [5] Wang S, et al. Segment-Level Attribution for Selective Learning of Long Reasoning Traces. ICLR, 2026.
>
> ### *Q1: Selection of ATPs*
>
> Please refer to ***W2***.
>
> ### *Q2: Hyperparameter Configurations*
>
> Please refer to ***W1***. In addition, jump prompts do not always induce a new reasoning perspective. Increasing $s$ generally fails to help the model escape thinking traps, resulting in **longer generations with little to no accuracy gain**.
>
> ### *Q3: Performance under Different Sampling Strategy*
>
> We conduct further experiments across several datasets and LRMs under a sampling setting. The results show that ASAG remains effective and is not limited to deterministic decoding.
>
> Please refer to the response to Reviewer gHML's ***W1*** for details.

---

> > ### Author Rebuttal · Reviewer_mbaZ · 2026-04-02
> >
> > Thank authors for the detailed response of the review. I think my concerns have been solved. I'll raise my score.

---

> > > ### Author Response · Authors · 2026-04-02
> > >
> > > Thank you for your careful review and constructive feedback. Your comments have helped us substantially improve the paper, particularly by clarifying the hyperparameter configurations, providing an analysis of ATP selection, and extending the evaluation to sampling settings. These revisions strengthen both the empirical support and the clarity of our method. We sincerely appreciate your insightful suggestions, which have made the revised paper more rigorous and easier to follow.

---

### Decision · Program_Chairs · 2026-04-30

**Decision:**

Accept (spotlight)

**Comment:**

The paper introduces ASAG, a training-free appraoch that infers the model's reasoning state and change the decoding strategy accordingly. It finds that when the model generates words such as "wait" and the attention becomes more distributed, further reasoning does not improve the final reasoning results. The authors design methods to let models come to an early decision with this observation. The reviewers acknolwedged the clear paper writing and solid experiment results. Further discussions in the rebuttal cleared reviewers' questions and concerns.